# The unfolded protein response reverses the effects of glucose on lifespan in chemically-sterilized *C. elegans*

Caroline Beaudoin-Chabot[1,5], Lei Wang [1,4,5], Cenk Celik [1], Aishah Tul-Firdaus Abdul Khalid [1], Subhash Thalappilly[1], Shiyi Xu[1], Jhee Hong Koh[1], Venus Wen Xuan Lim[1], Ann Don Low[1] & Guillaume Thibault [1,2,3] ✉

Metabolic diseases often share common traits, including accumulation of unfolded proteins in the endoplasmic reticulum (ER). Upon ER stress, the unfolded protein response (UPR) is activated to limit cellular damage which weakens with age. Here, we show that *Caenorhabditis elegans* fed a bacterial diet supplemented high glucose at day 5 of adulthood (HGD-5) extends their lifespan, whereas exposed at day 1 (HGD-1) experience shortened longevity. We observed a metabolic shift only in HGD-1, while glucose and infertility synergistically prolonged the lifespan of HGD-5, independently of DAF-16. Notably, we identified that UPR stress sensors ATF-6 and PEK-1 contributed to the longevity of HGD-5 worms, while *ire-1* ablation drastically increased HGD-1 lifespan. Together, we postulate that HGD activates the otherwise quiescent UPR in aged worms to overcome ageing-related stress and restore ER home-ostasis. In contrast, young animals subjected to HGD provokes unresolved ER stress, conversely leading to a detrimental stress response.

Diets of industrialised countries are enriched in processed carbohydrates and sugars that are rapidly converted into glucose, thus raising blood glucose levels promptly[1]. High blood glucose level is strongly associated with metabolic syndromes such as type II diabetes (T2D) and cardiovascular diseases consequently affecting lifespan[2,3]. First shown in young *Caenorhabditis elegans* adults[4,5] and subsequently, in other model organisms, a high-glucose diet (HGD) shortens lifespan[6,7]. In contrast, calorie restriction but not malnutrition has been previously shown to increase lifespan in rodents, and later, in other model organisms, including *C. elegans*[4,8]. Based upon these observations, the highly conserved insulin/insulin-like growth factor (IGF-1) signalling (IIS) and TOR signalling pathways modulate lifespan in several organisms[9–12]. However, the broader impacts of HGD on the lifespan of post-reproductive animals remain unaddressed as prior observations

on diet-related modulation of lifespan have mainly been limited to young adult animals.

Ageing is one of the most critical risk factors for the development of metabolic syndromes. Both T2D and insulin resistance have a strong association with endoplasmic reticulum (ER) stress. Upon ER stress, the unfolded protein response (UPR) is activated to limit cellular damage to adapt to stress conditions to restore ER homoeostasis. The UPR actuates multiple intracellular signalling pathways to restore ER and subsequently cellular homoeostasis. The UPR programme resolves ER stress by a transient decrease of translation, enhanced degradation of misfolded proteins and increased levels of ER-resident molecular chaperones[13–15]. However, the activation of genes through the IRE1 branch of the UPR tends to decrease, while the incidence of developing metabolic syndromes increases with age[16,17]. IRE1 has been shown to

[1]School of Biological Sciences, Nanyang Technological University, Singapore 637551, Singapore. [2]Mechanobiology Institute, National University of Singapore, Singapore 117411, Singapore. [3]Institute of Molecular and Cell Biology, A*STAR, Singapore 138673, Singapore. [4]Present address: Department Physiology and Biophysics, University of Miami Miller School of Medicine, Miami, FL 33136, USA. [5]These authors contributed equally: Caroline Beaudoin-Chabot, Lei Wang. ✉e-mail: thibault@ntu.edu.sg

extend lifespan in a cell-non-autonomous manner by remodelling lipid metabolism and promoting lipophagy[16,18,19]. Neuropeptide signalling originating from four glial cells is postulated to regulate ER stress resistance and longevity in *C. elegans*[20]. Although stress resistance correlates with increased longevity in several model organisms, the interplay between the UPR, HGD and longevity remain poorly understood.

Ageing is also characterised by the deterioration in physiological capacity and exacerbated environmental stress–resolving mechanisms that subsequently result in susceptibility to age-related diseases[21]. Several studies correlate ageing to oxidative damage incurred by reactive oxygen species (ROS)[22–24]. These reactive compounds damage the cell by changing lipid membrane properties, cross-linking cellular proteins and causing mutagenic changes to DNA. Oxidative and ER stress converge upon some forms of ROS that can activate the UPR[25,26]. The UPR mitigates the impairment induced by cellular stress[27]; however, the efficacy of the response declines with age[16,28]. A similar trend has been observed for other homoeostatic cellular stress responses[29,30], including autophagy[31] and the heat shock response[32] (HSR).

Here, we report that *C. elegans* fed a bacterial diet supplemented with 2% glucose at day 5 of adulthood (HGD-5) extend their lifespan compared to animals fed a normal diet (ND). In contrast, the lifespan of day one adult animals fed an HGD (HGD-1) was reduced, coinciding with previous reports[4,5]. A metabolic shift was observed in HGD-1 while glucose and infertility synergistically prolonged the lifespan of HGD-5, independently of DAF-16. In particular, we identified that UPR stress sensors ATF-6 and PEK-1 extended the longevity of HGD-5 animals, whereas IRE-1 exacerbated the decreased lifespan of HGD-1 worms. Based on these findings, we propose that HGD activates the otherwise quiescent UPR in aged animals to overcome ageing-associated stress and restore ER homoeostasis. In contrast, young animals subjected to high-glucose stress provokes unresolved ER stress, leading to a detrimental stress response.

## Results

### A high glucose diet extends the lifespans of aged adult animals

The shortening of lifespan by a HGD in young *C. elegans* and other model organisms is well-documented[4–7] but little is known on aged animals. To compare the effects of HGD on the lifespan of young adults and aged worms, wild type (WT) *C. elegans* of day 1 (D1) and day 5 (D5) of adulthood were fed a diet of UV-killed *Escherichia coli* OP50 bacteria supplemented with 2% glucose (HGD-1 and HGD-5, respectively). As previously reported[5], HGD remarkably shortened the lifespan of HGD-1 worms compared to the normal diet (ND) (Fig. 1a, Supplementary Data 1). In contrast, HGD-5 worms had a significantly extended lifespan compared to ND (Fig. 1b, Supplementary Data 1). As pharyngeal pumping declines with age[33] and with high-glucose diet[34], we asked whether glucose reduces bacterial uptake that may result in calorie restriction-induced longevity[4]. The pumping rate in D5 worms subjected to HGD for 24 h was comparable to ND (Fig. 1c). Subjecting D5 worms to 4 h starvation, however, increased pumping rate as previously reported[35]. To assess whether HGD influences food intake in D5 worms, we measured the change of OP50 density in liquid culture after 3 days incubation[36] (Fig. 1d). In WT worms, bacteria levels were significantly higher in HGD-1 while slightly higher in HGD-5 when compared to ND. Next, we measured total body glucose in aged WT worms fed either the ND or HGD (Fig. 1e). As expected, HGD significantly increased the total glucose levels in D5 worms compared to ND, coinciding with previous reports for young worms fed on HGD[5]. We then performed a motility assay to test if HGD-5 worms can access and uptake nutrients adequately. Wild-type adult worms supplied with HGD at day 5 for 24 h were more agile when compared to ND, while D1 worms fed on HGD were comparable to animals fed ND (Fig. 1f). To further assess the motility of HGD-5 worms, we carried out a thrashing assay on day 10 of adulthood and found an increased rate of body bending. Conversely, we observed mobility dysfunction in HGD-1 animals (Fig. 1g). Together, our findings suggest that HGD-5 worms live

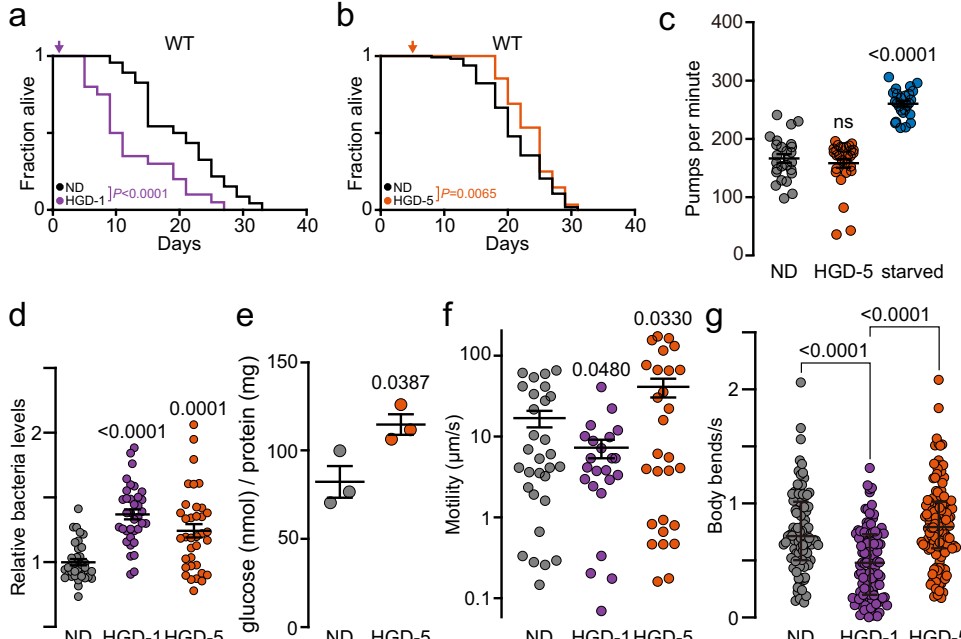

**Fig. 1 | Glucose extends lifespan of aged but not of young adult worms.**
**a** Lifespan assays of WT nematodes fed with UV-killed *E. coli* OP50 diet (normal diet, ND) alone or supplemented with 2% glucose (high glucose diet, HGD) at 1-day old (D1, HGD-1) and **b** 5-day old (D5, HGD-5) worms (NDa, *n* = 170; HGD-1, *n* = 90; NDb, *n* = 722; HGD-5, *n* = 772 including biological replicates). **c** Pharyngeal pumping rate (ND, *n* = 25; HGD-5, *n* = 28; starved, *n* = 25). **d** Bacteria intake relative to bacteria clearance within WT ND after 3 day incubation (ND, *n* = 417; HGD-1, *n* = 424; HGD-5,

*n* = 473), **e** Glucose levels inside of worms and **f** motility on bacteria-free nematode growth media (NGM) agar of worms (ND, *n* = 30; HGD-1, *n* = 23; HGD-5, *n* = 28) in D5 WT on 24 h ND, HGD or 4 h starvation. **g** Thrashing assay for WT, HGD-1 and HGD-5 worms at day 12. *P* values compared to WT on ND (ND, *n* = 91; HGD-1, *n* = 105; HGD-5, *n* = 130). *ns*, non-significant with *P* > 0.05. Data shown are the mean ± SEM. Statistical analysis was subjected to log-rank test for lifespan (**a**, **b**) or one-way ANOVA with Tukey's test (**c**–**g**).

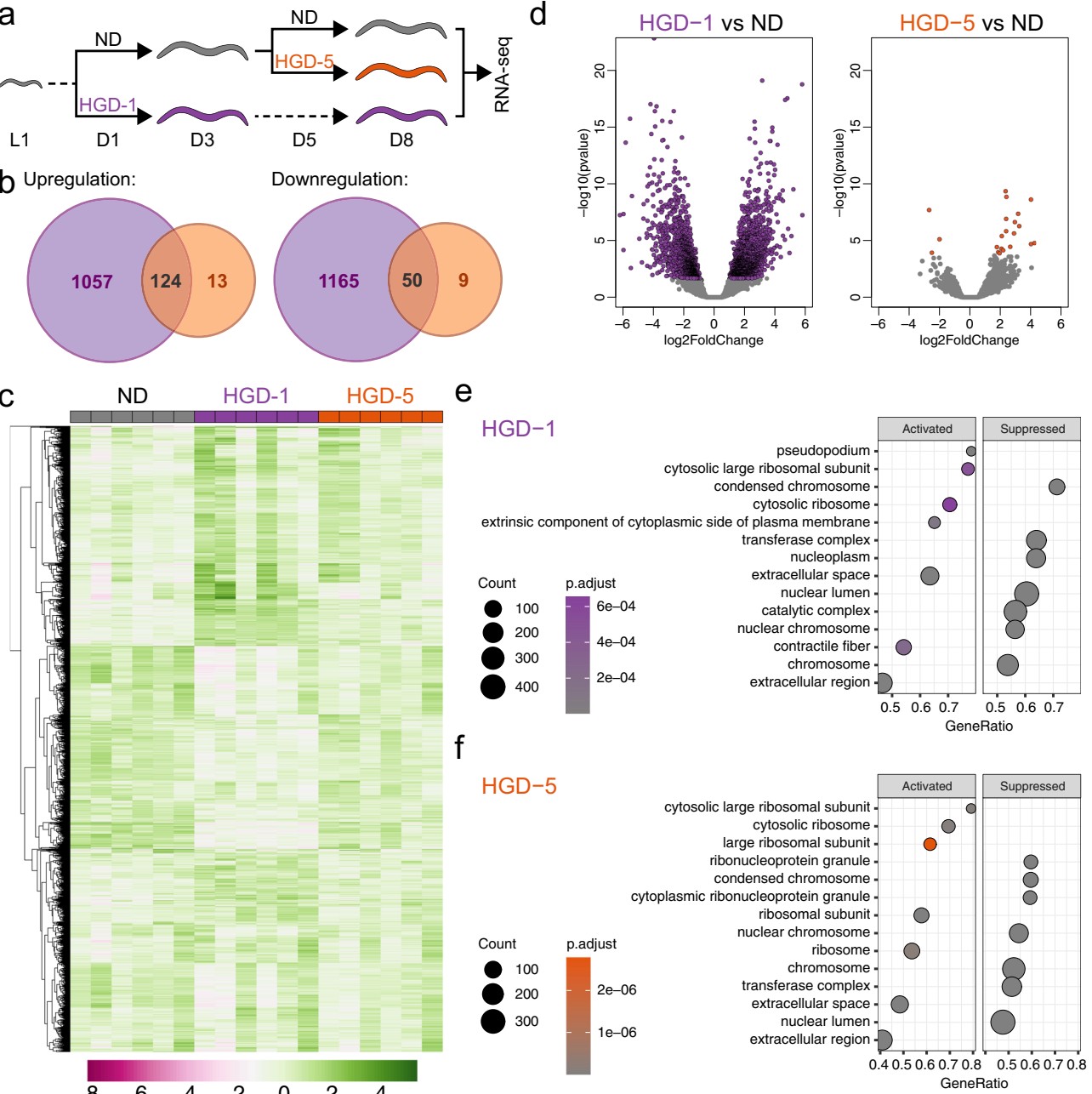

**Fig. 2 | HGD induces global transcriptomic changes in young but not aged animals. a** Experimental design of bulk RNA-seq. A single pool of synchronised worms was grown on ND until day 1 of adulthood (D1). From the same single pool, a subset of worms was fed on HGD at day 1 (HGD-1). The second subset of worms grown on ND was fed on HGD at day 5 (HGD-5) of adulthood (D5). Wrms fed ND, HGD-1, and HGD-5 were harvested at D8 (*n* = 6). **b** Venn diagram represents differentially expressed genes in HGD-1 and HGD-5 compared to ND. **c** Hierarchical clustering of the top 10,000 differently expressed genes in all samples. **d** Volcano plots of HGD-1 and HGD-5 compared to ND where coloured data points represent differentially expressed genes. Functional annotation analysis in HGD-1 (**e**) and HGD-5 (**f**).

longer and are vigorous with excess energy derived, at least in part, from glucose.

## HGD induces broader transcriptomic shifts in young than aged animals

Quantitative bulk RNA sequencing was conducted for WT animals to understand how HGD extends the lifespan of aged animals (Fig. 2a). A single pool of synchronised worms was grown on ND to day 1 of adulthood (D1). From the same pool of worms, a subset of worms was fed HGD on day 1. The second subset of worms fed ND was subjected to high glucose at day 5 (HGD-5) of adulthood (D5). Worms fed ND,

HGD-1 and HGD-5 were harvested eight days later (Supplementary Fig. 1). Overall, HGD-1 induced a global transcriptional response while the changes in regulated genes were more targeted in HGD-5 (Fig. 2b, deposited in GEO Accession number GSE182981). Compared to ND, 124 and 50 genes were upregulated and downregulated, respectively, in both HGD-treated animals. Hierarchical clustering of the top 10,000 differently expressed genes in all samples and volcano plots show that HGD-5 was comparable to ND while sharing some common clusters with HGD-1 (Fig. 2c, d). To interrogate how HGD induces cellular changes, we performed gene set enrichment analyses and reported the top 14 categories (Fig. 2e, f). Major pathways suppressed by HGD

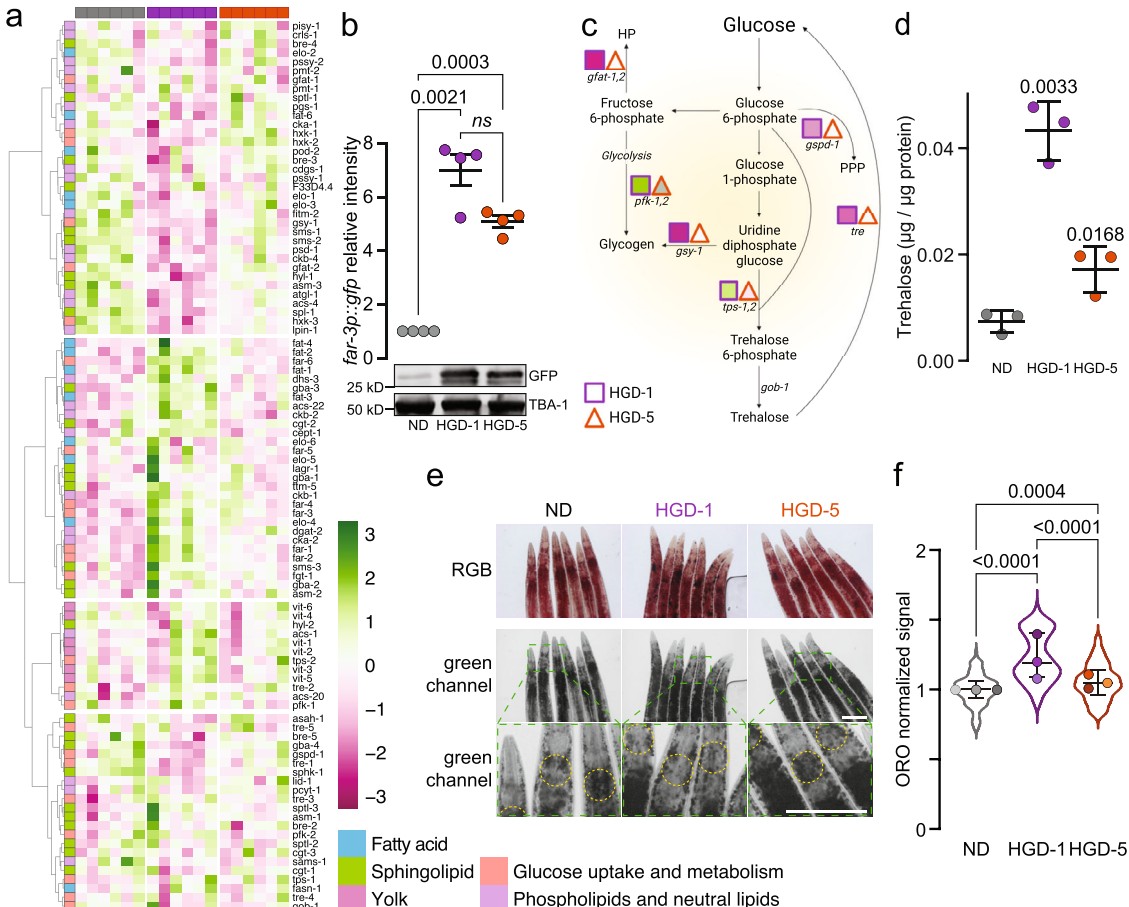

**Fig. 3 | The metabolic landscape is differently modified in young and aged animals fed HGD. a** Heat maps of glucose and lipid metabolism genes. **b** Quantification and representative immunoblot of GFP levels in day 10 *far-3p::gfp* animals fed ND, HGD-1 and HGD-5 (*n* = 4). **c** Major glucose metabolic pathways in *C. elegans*, f6p fructose-6-phosphate, g1p glucose-1-phosphate, g6p glucose-6-phosphate, HP hexosamine pathway, PPP pentose phosphate pathway, UDP-G uridine diphosphate glucose. Adapted from Seo et al.[88] and created with BioRender.com. **d** Trehalose levels of worms treated as in WT, HGD-1 and HGD-5 at day 10 (*n* = 3).

**e** Representative images of Oil Red O (ORO) staining of ND, HGD-1 and HGD-5 animals harvested on day 10. Lipid droplets were quantified from the green channel within regions of interest (yellow-dashed circles) as previously reported[55]. Scale bar represents 100 μm. **f** Quantification of ORO-stained lipid droplets at the pharynx of animal as in **e** (ND, *n* = 127; HGD-1, *n* = 110; HGD-5, *n* = 112 including biological replicates). Data shown are the mean ± SEM. Statistical analysis was subjected to one-way ANOVA with Tukey's test.

regardless of the time of supplementation include condensed chromosome, transferase complex, nuclear lumen and nuclear chromosome, while activated systems include cytosolic large ribosomal subunit, cytosolic ribosome and extracellular space and region. The data suggest that HGD suppresses transcriptional programmes in the nucleus while focusing on the translational processes, indicating the need of metabolic pathways for longevity.

**The metabolic landscape is divergent in young and aged animals fed HGD**

Dietary glucose molecules are converted into energy and metabolites intracellularly while the excess is transformed into storage molecules for needs thereafter. We further examined glucose and lipid metabolic genes to characterise glucose intake and metabolism in aged animals fed HGD (Fig. 3a). A subset of genes related to fatty acids, phospholipids, neutral lipids and sphingolipid metabolism, as well as yolk regulation, glucose metabolism and storage, were differentially regulated in HGD-1 and, to a smaller extent, in HGD-5 compared to ND. For instance, fatty acid/retinol-binding protein (*far*) genes were profoundly upregulated in HGD-1 and have been reported to increase in response to glucose[37]. To validate this finding, we monitored the GFP levels in day ten *far-3p::gfp* animals by immunoblot. We observed a notable increase in GFP levels of HGD-1 and HGD-5 compared to ND animals

(Fig. 3b). Findings indicate that intracellular glucose levels are similar in HGD-1 and HGD-5 despite being fed HGD for 9 and 5 days, respectively.

Next, we examined metabolites related to glucose metabolism and storage and found that HGD-1 animals stored more trehalose than HGD-5 and ND, the latter having stored the least (Fig. 3c, d). We further quantified fatty acid (FAs) levels as excess glucose can prompt fatty acid remodelling (Supplementary Fig. 2). Levels of myristic acid (C14:0), palmitic acid (C16:0) and palmitoleic acid (C16:1) were more pronounced in both HGDs compared to ND animals. An overall increase in polyunsaturated fatty acids (PUFAs) was also observed in HGD-5, whereas γ-linolenic acid (C18:3n6) was exclusively lower than in ND animals. Together, these changes in FAs correlate with the expression levels of corresponding FA synthesis genes (Fig. 3a). As anticipated, we observed an increased number of neutral lipid storage organelles, namely lipid droplets, in HGD-1 and HGD-5 compared to ND, while HGD-1 contained the highest number (Fig. 3e, f). Together, data suggest that HGD has a substantial impact on glucose metabolism and storage in young animals.

**Glucose and infertility synergistically prolong lifespans of aged animals**

One of the main differences between young and aged animals is that HGD-1 and HGD-5 are subjected to glucose pre- and post-reproduction,

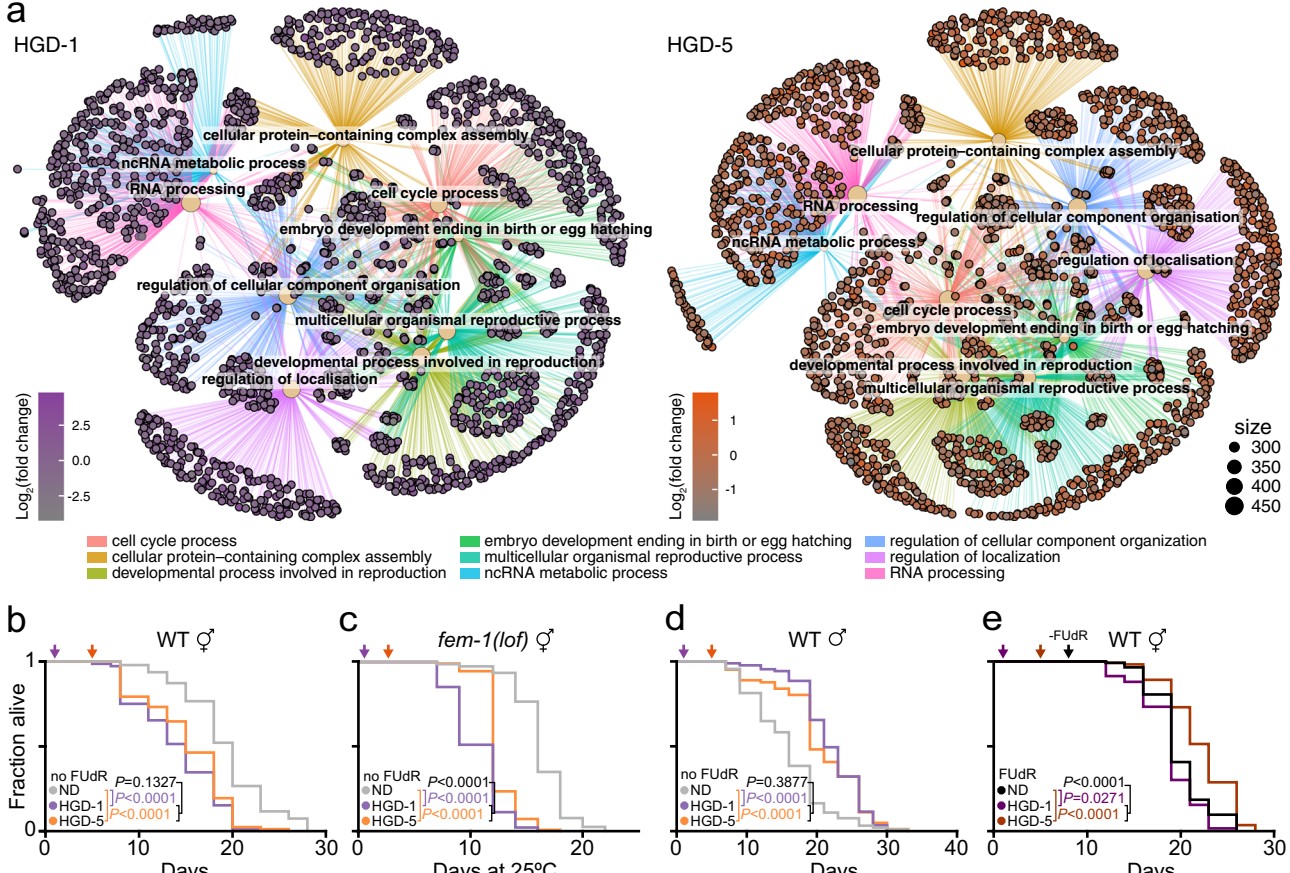

**Fig. 4 | Glucose and infertility synergistically drive the lifespan of aged animals.** **a** Cnetplots depict the linkages of genes and biological concepts in HGD-1 and HGD-5 compared to ND, respectively, where each nod shows the most enriched pathways. **b-c** Lifespan assays of WT (ND, n = 287; HGD-1, n = 255; HGD-5, n = 255 including biological replicates) (**b**) or *fem-1(lof)* hermaphrodite (ND, n = 186; HGD-1, n = 191; HGD-5, n = 209 including biological replicates) (**c**) animals fed normal diets (ND) or high glucose diet (HGD) from 1- (HGD-1) or 5-day old (HGD-5) in the absence of 5-Fluoro-2′-deoxyuridine (FUdR). **d** Lifespan assays of WT male animals fed ND, HGD-1, or HGD-5 in the absence of FUdR (ND, n = 150; HGD-1, n = 139; HGD-5, n = 143 including biological replicates). **e** Lifespan assays of WT hermaphrodite animals fed on ND, HGD-1, HGD-5 in the presence of FUdR from L4 to D8 (ND, n = 213; HGD-1, n = 216; HGD-5, n = 208 including biological replicates). Data shown are the mean ± SEM. Statistical analysis was subjected to log-rank test for lifespan.

respectively. As we identified a shift in lipid metabolism and storage that play a role in embryogenesis[38], we sought to analyse the RNA-seq data. In both HGD-1 and HGD-5, major systems related to reproduction were upregulated (Fig. 4a). These findings prompted us to explore the synergy between glucose and fertility in young and aged animals.

Five-fluoro-2′-deoxyuridine (FUdR) is a commonly used reagent for chemically sterilising adult worms in lifespan assays[39]. FUdR inhibits germline stem cell proliferation and the production of intact eggs in adults[40]. However, FUdR alone has been shown to extend lifespan by promoting stress responses in mutant animals or during stressing growth conditions[41–45]. Therefore, we asked if there is a synergic effect between FUdR and HGD in promoting the longevity of HGD-5 animals. In the absence of FUdR, the lifespan of WT animals on ND was comparable to those grown in the presence of FUdR (Fig. 4b, Supplementary Data 1), in agreement with previous reports[43,44]. Similarly, animals on HGD-1 exhibited a shorter lifespan compared to ND. Conversely, the lifespan of HGD-5 animals was drastically shortened in the absence of FUdR and comparable to the lifespan of animals on HGD-1. To further assess the role of fertility in modulating lifespan, we used long-lived germline defective *fem-1(loss-of-function; lof)* animals[46]. In contrast to FUdR which prevents cell division indiscriminately when subjecting stage 4 larval animals, *fem-1* mutation results in a loss of stem cells, which are precursors to gamete cells[47]. As expected, *fem-1(lof)* animals lived longer than WT on ND at 25 °C (Supplementary Fig. S3a and Table 1). Surprisingly, the lifespan of *fem-1(lof)* animals fed

on HGD, at the equivalent of day 5 at 25 °C, was significantly shorter compared to ND in the absence of FUdR (Fig. 4c, Supplementary Data 1). Similarly, aged germline defective *glp-1(lof)* animals exhibited a shorter lifespan on HGD (Supplementary Fig. S3b and Table 1). Together, these findings suggest that germline proliferation plays a role in regulating the longevity of HGD-5 animals.

To further dissect HGD-induced longevity to the role of the germline in modulating lifespan, we performed lifespan assays in male worms. The lifespan of male WT animals fed ND was mildly reduced by FUdR (Supplementary Fig. 3c and Table 1). Next, we monitored the lifespan of WT males fed HGD in the absence of FUdR (Fig. 4d, Supplementary Data 1). As previously reported[48], the lifespan of male WT worms fed on HGD-1 was extended compared to ND while remaining comparable to the lifespan of HGD-5. In the presence of FUdR, lifespans of either HGD-1 or HGD-5 males were still extended compared to ND (Supplementary Fig. 3d and Table 1). Together, these findings suggest that HGD promotes longevity in young and aged male worms independently of FUdR.

By convention, FUdR is introduced to the animals from larva stage 4 (L4) and supplemented throughout adulthood[49,50]. However, *C. elegans* lays eggs about the first four days of adulthood[51]. Therefore, we interrogated if FUdR affects the longevity of animals beyond the fertility window. WT animals fed on ND, HGD-1 or HGD-5 were exposed to FUdR from L4 to day 7 of adulthood (Fig. 4e, Supplementary Data 1). The lifespan of HGD-1 animals was slightly shorter than ND, while HGD-

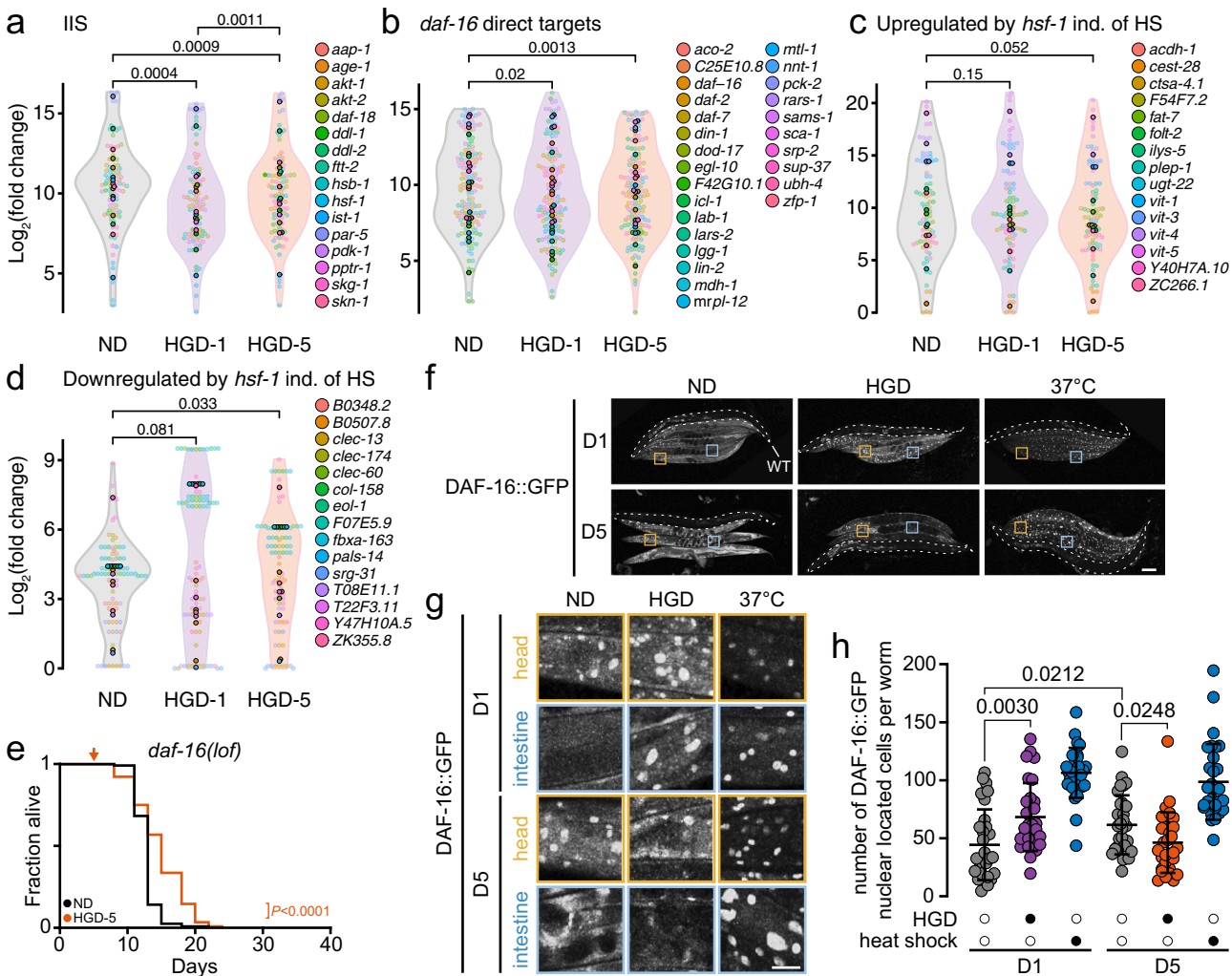

**Fig. 5 | Glucose extends lifespan in aged worms independently of the DAF-16 (FOXO).** Differentially regulated genes downstream of the insulin/insulin-like growth factor signalling (IIS) (**a**), daf-16 (**b**), hsf-1 independently (ind.) of heat shock (HS) (**c**, **d**) in animals fed on the normal diet (ND) or high-glucose diet (HGD) from 5-day old (HGD-5). **e** Lifespan assays of *daf-16(lof)* animals fed on the ND or HGD-5 (ND, *n* = 502; HGD-5, *n* = 536 including biological replicates). **f**, **g** Representative images of DAF-16::GFP localisation in 1- (D1) and 5-day old (D5) worms on 16 h ND, HGD (filled circles) or 1 h heat shock (filled circles) at 37 °C (*n* = 60 for each condition including biological replicates). Scale bars represent 100 μm (f) and 25 μm (g). Highlighted worms with dashed lines are WT animals. **h** Quantification of GFP in panel f (*n* = 30). *P* values to ND of WT. Data shown are the mean ± SEM. Statistical analysis was subjected to one-way ANOVA with Tukey's test (**a**–**d**, **h**) or log-rank test for lifespan (**e**).

5 animals lived longer compared to ND. The number of eggs laid per HGD-1 animal was slightly reduced compared to ND animals in the absence of FUdR at day 4 of adulthood (Supplementary Fig. 3e). Interestingly in the presence of FUdR, the number of eggs laid per HGD-1 animal was dramatically reduced compared to ND animals. Together with the lifespan assays in the absence or presence of FUdR, findings suggest that HGD in young animals decreases both the number of laid eggs and lifespan.

### Glucose-induced longevity of aged worms is driven independently of DAF-16/FOXO

The highly conserved IIS pathway modulates growth, differentiation, metabolism from nutrient availability and fluctuations in environmental conditions[9–12,52]. The activation of the IIS pathway reduces the lifespan of HGD-1[5]. DAF-2-modulated transcription factor DAF-16 [orthologue of forkhead box O (FOXO)], a member of the IIS pathway, increases the levels of carbon storage molecules including trehalose, and triglycerides[53–60] which were both predominantly stored in HGD-1 (Fig. 3c–f). Consequently, we assessed gene expression of the IIS pathway and DAF-2-regulated transcription factor DAF-16 (Fig. 5a, b).

Overall, more genes were upregulated by DAF-16 in HGD-1 than HGD-5, suggesting that DAF-16 might not modulate HGD-5 lifespan. The transcription factor HSF-1 is another important regulator of longevity which is regulated by the IIS pathway independently of heat shock[61,62]. In general, HSF-1 target genes were significantly upregulated and downregulated in HGD-1 animals while only downregulated genes were significantly different in HGD-5 compared to ND (Fig. 5c, d). To determine if this pathway modulates the longevity of HGD-5 animals, we carried out the lifespan assay of short-living *daf-16(lof)* animals. The lifespan of *daf-16(lof)* on HGD-5 was more prolonged than ND (Fig. 5e, Supplementary Data 1), suggesting that the lifespan extension is independent of DAF-16. Next, to quantify the activation of DAF-16 in D5 worms, we monitored DAF-16::GFP localisation (Fig. 5f–h). In D1 worms fed on HGD for 16 h, we observed a remarkable increase in nuclear localisation of DAF-16::GFP, indicative of a HGD-induced stress response. In contrast, the number of DAF-16::GFP-localised nuclei in HGD-5 was lower compared to ND. As expected, DAF-16::GFP profoundly colocalised into nuclei of HGD-1 and HGD-5 worms upon heat shock[63], showing DAF-16 activation is not compromised by ageing. Together, these findings suggest that DAF-16 is excluded from the

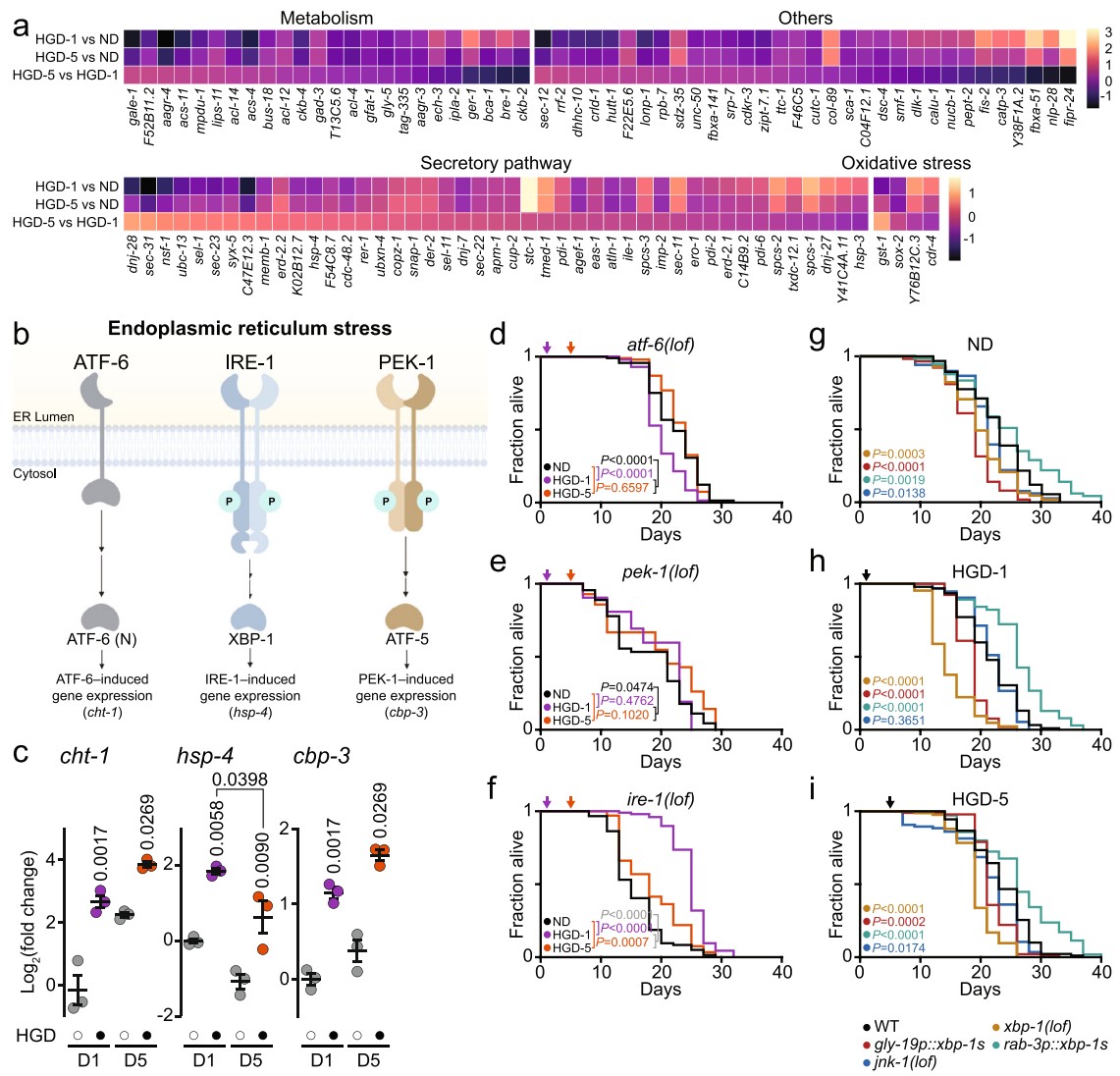

**Fig. 6 | The UPR responsiveness is remodelled in aged worms extending life-span upon HGD. a** Differentially expressed genes related to the UPR comparing each diet group with one another. **b** Overview of the unfolded protein response (UPR) in *C. elegans*. Created with BioRender.com. **c** qPCR comparing expression of genes *cht-1*, *hsp-4*, and *F40F12.7* induced from UPR branches ATF-6, IRE-1, and PEK-1, respectively, in 1- and 5-day-old WT worms fed 24 h normal diet (ND, open circles) or high glucose diet (HGD, filled circles). Data shown are the mean ± SEM (*n* = 3). **d–g** Lifespan assay of 1- and 5-day-old *atf-6(lof)* (ND, *n* = 287; HGD-1, *n* = 99; HGD-5,

*n* = 280 including biological replicates) (**d**) and *pek-1(lof)* (ND, *n* = 215; HGD-1, *n* = 52; HGD-5, *n* = 104 including biological replicates) (**e**) worm mutants fed ND or HGD. **f** Lifespan assay of *ire-1(lof)* worm mutant treated as in **d** (ND, *n* = 229; HGD-1, *n* = 387; HGD-5, *n* = 263 including biological replicates). Lifespan assays of WT, *xbp-1(lof)*, *gly-19p::xbp-1s*, *rab-3p::xbp-1s* and *jnk-1(lof)* fed ND (**g**), HGD-1 (**h**) or HGD-5 (**i**). Statistical analysis was subjected to one-way ANOVA with Tukey's test (**c**) or log-rank test for lifespan (**d–i**).

nucleus and is not involved in extending the lifespan of D5 animals fed HGD.

## IRE-1 modulates the longevity of animals on HGD

The UPR is activated to limit cellular damage to adapt to stress conditions and restore ER homoeostasis, while an acute UPR activation drives pro-apoptotic pathways[13]. As high glucose has been reported to induce ER stress[64,65], we analysed our RNA-seq data to identify UPR target genes that are differentially regulated. Several UPR target genes were upregulated in HGD animals compared to ND animals (Fig. 6a). Among these, *stc-1*, an orthologue of human *HSPA13*, and *fipr-24*, an uncharacterised protein, are both modulated by IRE-1 as well as signal peptidase complex subunit *sec-11*, an orthologue of human *SEC11C*, genes were strongly upregulated upon HGD. Overall, more UPR-mediated genes were upregulated in HGD-5 when compared to HGD-1 animals. Notably, secretory pathway genes were highly expressed in HGD-5 animals including *ubc-13*, an orthologue of human *UBE2N*, *sel-1*,

an orthologue of human *SEL1L*, *cdc-48.2*, an orthologue of human *VCP*, *ubxn-4*, an orthologue of human *UBXN4*, *der-2*, an orthologue of human *DERL2*, and *sel-11*, an orthologue of human *HRD1*, genes which are implicated in the ER-associated degradation (ERAD). This suggests that unfolded proteins at the ER are likely more efficiently cleared in HGD-5 animals.

To validate if HGD induces the UPR in young and aged *C. elegans*, we measured the activation of the UPR in animals fed HGD for 24 h. UPR target genes *cht-1*, *hsp-4* and *cbp-3* were selected to monitor the activity of ATF-6, IRE-1 and PEK-1, respectively (Fig. 6b and Supplementary Fig. 4a–c). Feeding 1- and 5-day-old WT worms for 24 h on HGD-induced ER stress with significant activation of the UPR through the three ER stress sensors (Fig. 6c). However, HGD-induced upregulation of *hsp-4* through the IRE-1 branch was significantly lower in D5 compared to D1 worms, consistent with previous findings[16]. On the other hand, the upregulation of *cht-1* and *cbp-3* from HGD in D5 were comparable to D1 worms. These findings suggest that ATF-6 and PEK-1

might compensate for the weak activation of IRE-1 in D5 animals fed a ND, consequently playing an important role in maintaining ER homoeostasis in aged animals. Hereafter, we asked if the UPR plays a role in extending the lifespan of HGD-5 animals. The lifespans of *atf-6(lof)* and *pek-1(lof)* animals on HGD from D1 or D5 were similar to ND (Fig. 6d, e, Supplementary Data 1). These findings indicate that both ATF-6 and PEK-1 participate in extending the longevity of D5 worms on HGD. In contrast, HGD extended the lifespan of D5 *ire-1(lof)* animals when compared to ND (Fig. 6f, Supplementary Data 1). Surprisingly, the lifespan of *ire-1(lof)* animals on HGD-1 was longer than *ire-1(lof)* animals on ND and HGD-5 as well as WT animals on ND (Supplementary Fig. 4d and Table 1).

To further characterise the role of the IRE-1 pathway in modulating the longevity of animals subjected to HGD, we conducted lifespan assays in several mutant worms subjected to ND, HGD-1 and HGD-5. Upon ER stress, IRE-1 activation causes increased splicing of *xbp-1* mRNA (*xbp-1s*), resulting in the translation of the transcription factor XBP-1 which regulates a subset of UPR genes[15]. We included *xbp-1(lof)* animals as well as mutant animals with constitutive expression of *xbp-1s* in intestinal (*gly-19p::xbp-1s*) and neuronal (*rab-3p::xbp-1s*) cells[16]. A constitutive expression of *xbp-1s* in specific tissues translates into the upregulation of XBP-1 target genes regardless of ER stress. As an alternative to the IRE-1/XBP-1 axis, we also incorporated *jnk-1(lof)* animals. IRE-1 exhibits pro-apoptotic properties by phosphorylating c-Jun amino-terminal kinase (JNK-1) during chronic ER stress. As previously reported[20], the lifespan of *xbp-1(lof)* animals was shorter compared to WT on ND (Fig. 6g). The lifespan of *gly-19p::xbp-1s* animals was shorter while *rab-3p::xbp-1s* animals exhibited a significant increase in longevity when compared to WT on ND, similar to previous reports[16,19]. Finally, the lifespan of *jnk-1(lof)* animals on ND was slightly shorter compared to WT, validating previous findings[66,67]. The decreased lifespan of *xbp-1(lof)* animals was exacerbated upon HGD-1 when compared to WT animals on HGD-1 while HGD-5 failed to extend the lifespan of *xbp-1(lof)*, similar to *ire-1(lof)* (Fig. 6h, i). This suggests that XBP-1 is necessary to extend the lifespan of aged animals on HGD while IRE-1 might hyperactivate the UPR through *xbp-1s* in certain tissues. In agreement with this hypothesis, the constitutive expression of *xbp-1s* in neuronal cells increased the lifespan of *rab-3p::xbp-1s* animals on HGD-1 and to a less extent on HGD-5 compared to ND. On the other hand, the constitutive expression of *xbp-1s* in intestinal cells had little effect on the lifespans of *gly-19::xbp-1s* animals on HGD when compared to WT animals. Similarly, *jnk-1(lof)* animals exhibit similar lifespans on the different diets in comparison to WT animals. Together with the transcriptomic analysis, these findings suggest that IRE-1 exacerbates the decreased lifespan of HGD-1 animals, while ATF-6 and PEK-1 promote longevity of HGD-5 animals.

## Discussion

Ageing is defined as a progressive loss of physiological functions, including reduced fertility, decreased protein homoeostasis, cellular senescence, genomic instability and disrupted metabolic homoeostasis[22]. Particularly relevant, infertility has been shown to promote longevity in a DAF-16-dependent manner using the *C. elegans glp-1(lof)* mutant that shows a complete loss of germ cells[68]. Endocrine signalling and multiple transcriptional pathways increase lifespan in gonad-ablated animals[69]. In addition, an increase in insulin signalling through HGD promotes germline proliferation[70]. Thus, germline proliferation in young adult worms combined with HGD could yield a synergistic effect, resulting in accelerated ageing. In contrast, HGD introduced at the post-reproductive stage could stimulate age-attenuated cellular stress responses, consequently extending lifespan through the restoration of cellular homoeostasis. A severe decline in proteostasis correlates with the termination of germline proliferation[71]. Our study points to a specific life stage from which it could be beneficial to promote artificial proteostasis and

subsequently attenuate the consequences of ageing. Interestingly, animals subjected to HGD at the pre-reproductive stage or short HGD exposure in young adults have been reported to extend lifespan[72,73]. The timing and length for intervention might be critical, as it could exacerbate ageing if done unduly early and persistently. It remains to be interrogated whether the treatment that activates the UPR in post-reproductive mammals correlates with an extension of lifespan through the clearance of intracellular damage.

Extended lifespan is coupled with increased motility and energy metabolism[74,75], indicating healthy ageing. We found that post-reproductive worms were metabolically more active on HGD by displaying higher mobility, increasing total ATP and oogenesis, thereby suggesting healthier ageing than worms fed on the normal diet. As the proteostasis network declines during ageing, proper digestion of bacteria—including pharyngeal grinding, intestinal lysozymes, saposins and amebapores—might be impaired in aged *C. elegans*[76]. Evidence suggests that the transcriptome and proteome related to glucose metabolic pathways differ in young and post-reproductive worms[77,78]. Similarly, in mice, metabolites derived from glucose, fatty acids and amino acids decline dramatically upon ageing[79]. One could hypothesise that higher levels of HGD-induced insulin production are essential in stimulating the IIS pathway while its responsiveness deteriorates with age. Consequently, it is reasonable to conceive that the glucose diet will restore healthy calorie intake in aged animals while becoming exorbitant when combined with well-digested bacteria in young animals.

The highly conserved insulin signalling pathway promotes lifespan extension when attenuated or muted in *C. elegans* and other multicellular organisms[80]. Ablation of *age-1* or *daf-2* genes doubles the lifespan of *C. elegans*, while the absence of the FOXO transcription factor DAF-16 drastically expedites tissue ageing; hence, reducing longevity[9,63,81]. A HGD is sufficient to shorten longevity by downregulating DAF-16 activity through DAF-2 activation in young adults[5]. Similarly, some dietary restriction protocols increase lifespan by upregulating DAF-16 activity, increasing cellular stress surveillance[82,83]. Surprisingly, our findings suggest that while the ability to activate DAF-16 remains stable until the fifth day of adulthood, it is not involved in promoting the lifespan of post-reproductive worms subjected to HGD. It is, therefore, reasonable to conclude that DAF-16 is inactivated through DAF-2 which promotes glucose uptake and metabolism[84]. In addition, DAF-2 may be necessary through crosstalk with the UPR programme as the IRE-1/XBP-1 axis has been previously reported to compensate for the ablation of the IIS pathway[85]. Interesting, our findings suggest that neuronal XBP-1 is necessary to extend the lifespan of animals subjected to HGD from young adults (Fig. 6h). It would be interesting to elucidate if a synergy between DAF-2 and XBP-1 in neuronal cells is necessary to promote longevity of animals fed HGD.

Preventing decreased protein quality control is imperative for maintaining cellular homoeostasis, whose surveillance takes place in the ER by the UPR pathway[86]. Defects in the pathway compromise the proteostasis network in the ER and promote the accumulation of damaged and misfolded proteins that could be accelerated further during ageing[77]. ER-localised protein disulphide isomerases (PDIs), chaperone calreticulin (CRT-1), HSP-4 paralogue and HSP-3 (Hsp70 orthologue) have been reported to decline by about two-fold throughout *C. elegans* lifespan. Similarly, ER stress-induced upregulation of HSP-4 through the IRE-1 branch of the UPR is attenuated drastically during ageing[16]. On the other hand, the downstream target of the UPR sensor IRE-1, XBP-1, has been shown to act conjointly with DAF-16 to enhance ER stress resistance and thus longevity of IIS mutant worms[85]. Subsequently, the intervention of the UPR in maintaining ER and cellular homoeostasis is critical to extending the lifespan. In contrast, our findings suggest that IRE-1 is detrimental to the longevity of young animals on HGD (Fig. 6f). We hypothesise that HGD-driven hyperactivation of IRE-1 in young animals may induce premature cell

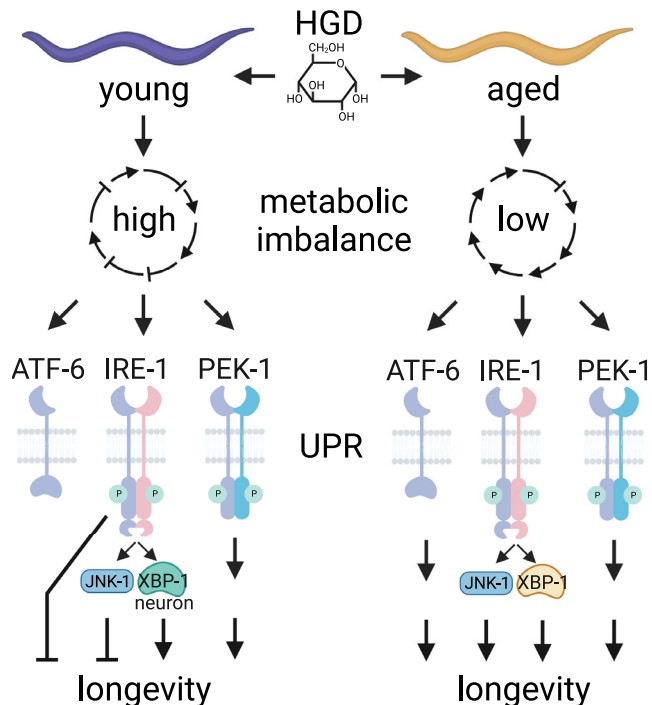

**Fig. 7 | HGD-induced UPR modulates longevity differently in young and aged animals.** *C. elegans* fed high glucose diet (HGD) at a young (day 1 of adulthood) or older (day 5 of adulthood) age induces a high and low metabolic imbalance, respectively. In turn, HGD reduces the longevity of young animals in an IRE-1-dependent manner while promoting an increase in lifespan via the ER stress sensors ATF-6 and PEK-1 in aged animals. The proposed model is applicable to chemically-sterilized worms. Created with BioRender.com.

death. However, the relative function of the ATF-6 and PEK-1 branches of the UPR is largely unknown during ageing. Our findings suggest the decline in IRE-1-mediated UPR responsiveness in aged worms compromises cellular homoeostasis[77] while ER stress agents such as glucose hyperactivate IRE-1 in young animals (Fig. 7). Additionally, the highly conserved UPR sensors ATF-6 and PEK-1 compensate for the decline of IRE-1 in aged animals to promote longevity upon HGD. Future studies should identify drugs that will selectively activate the ATF-6 and PEK-1 pathways while attenuating the IRE-1 pathway without leading to an acute UPR activation. Ideally, the fitness of the UPR should persist throughout life, in particular, during older age to prevent the age-related accumulation of cellular damage[87] and consequently premature death.

## Methods

### *C. elegans* strains, RNAi constructs and bacterial strains

All strains were grown at 20 °C using standard *C. elegans* methods as previously described[89,90]. Nematode growth media (NGM) agar plates were seeded with *E. coli* strain OP50 (UV-killed bacteria for lifespan assays) for normal growth and with HT115 bacteria for RNAi feeding. NGM agar plates were supplemented with 2% glucose (high glucose diet, HGD) when indicated. RNAi feeding was performed as previously described[91], and the RNAi library was obtained from the Fire lab[92]. The plasmids were verified by sequencing. *C. elegans* strains wild type N2, *atf-6(ok551)*, *daf-16(mu86)*, *daf-16::GFP(tj356)*, *fem-1(hc17)*, *glp-1(e2144)*, *ire-1(ok799)*, *jnk-1(gk7)*, *pek-1(ok275)*, *rab-3p::xbp-1s(uthIs270)*, *xbp-1(zc12)* and bacteria strains *OP50* as well as *OP50-GFP* were gifted from the *Caenorhabditis* Genetics Centre. *far-3p::GFP(bc14852)* was a gift from Seung-Jae Lee laboratory[37]. *gly-19p::xbp-1s(uthIs388)* was generously provided by Rebecca Taylor laboratory[19].

### Lifespan assay on solid plates

Lifespan assays were performed at 20 °C as previously described[49]. Animals were transferred to NGM plates containing 2% glucose on day 1 or 5 of adulthood when indicated. Day 1 was defined as the egg hatching event. Pyrimidine analogue 5-fluoro-2′-deoxyuridine (FUdR, Sigma; St. Louis) was supplemented with a concentration of 50 μM to L4 stage worms to prevent the development of progeny. Adults were scored manually as dead or alive every 2-3 days. Nematodes that ceased pharyngeal pumping and had no response to gentle stimulation were recorded as dead. Worms that were dead due to desiccation were excluded from the analysis.

### Lifespan assay in liquid culture

Around 10 synchronised L1 stage WT worms were seeded into a well of a 96-well microplate containing 150 μl of S-medium supplemented with 1 mg/mL of UV-killed OP50, 50 μg/mL Carbenicillin, 0.1 μg/mL Amphotericin B[90]. Upon reaching young adult, FUdR was added to a final concentration of 0.12 mM. Two per cent D-glucose was added to the respective wells on Day 5. Culture media was changed every 2 days. Worms were counted and scored daily using an inverted microscope (CKX53, Olympus, Japan) using a 4× or 10× objective until the death of all worms was recorded. Intense lighting was used to induce animal movement to determine survival. Wells that contained hatched eggs or contamination were censored from the analysis.

### Pharyngeal pumping assay

Pharyngeal pumping was measured by monitoring the number of pharyngeal bulb contractions in a 60-s interval. Videos were recorded with a Dino-eye Eyepiece camera (Dino-eye, Taiwan) fitted onto a stereomicroscope (Nikon, Tokyo) using DinoCapture 2.0. Contractions were carefully monitored through videos at 0.4× playback speed. At least 20 worms were recorded for each group. Worms that exhibited under 5 contractions per 10 s were considered as not pumping and excluded from the analysis.

### Quantification of bacteria intake

The consumption of *E. coli* OP50 by worms was assessed in 96-well plates as previously described[36]. Each well contained ~10 5-day-old worms and 0.37 $OD_{600}$/ml bacteria in 50 μl of S-complete liquid medium supplemented with 50 μg/ml carbenicillin and 50 μM FUdR. After 72 h incubation, plates were shaken at 200 rpm for 15 min and the absorbance at 600 nm was measured. The relative number of bacteria consumed was obtained from the difference in the absorbance at 600 nm (Tecan Infinite M200Pro with i-control software) normalised to the number of worms. The same drug-supplemented media without worms was set as blank. At least 157 nematodes were recorded for each group. Bacteria intake was calculated from the difference in absorbance over the 72 h period.

### Glucose assay

Glucose levels were measured as previously described[5] with some modifications. Approximately 1000 worms were harvested following 24 h growth on NGM supplemented with 2% glucose when indicated. Pelleted worms were lysed by bead beating in RIPA buffer (50 mMTris-HCL,150 mM NaCl, 2 mM EDTA, 1%NP-40, 0.1% SDS) and spun down at $12,000 \times g$ for 5 min, 4 °C to remove the debris. The clarified lysate was transferred to a new tube and protein content was determined by bicinchoninic acid (BCA) assay. Trichloroacetic acid (TCA) was added to the remaining lysate to a final concentration of 10% and spun down at $12,000 \times g$ for 5 min, 4 °C. The supernatant was neutralised to pH 7 with 1 N NaOH containing 100 mM potassium phosphate. Glucose levels were measured using the Amplex Red Glucose/Glucose Oxidase kit (Thermo Fisher, Waltham, MA) following the manufacturer's protocol using Tecan Infinite M200Pro with i-control software.

## Thrashing assay

Synchronised worms were harvested at D8 and washed three times with M9. The number of worms was adjusted in M9 medium to 10 worms per 10 µl. The thrashing of the worms was recorded 1 min at ×0.8 magnification of an SMZ1500 Nikon Stereomicroscope. Further analysis was done with the wrMTrck plugin in Fiji software version 1.53c. Student's *t*-test was performed using GraphPad Prism version 9.2.0 for Windows (GraphPad Software, Inc., San Diego, CA).

## Bulk RNA sequencing and data processing

A single pool of ~6000 synchronised worms was grown on ND to day 1 of adulthood (D1). From the same single pool, a subset of ~2000 worms was fed HGD-1. The second subset of ~2000 worms grown on ND was fed HGD-5 at day 5 of adulthood (D5). Worms fed ND, HGD-1, and HGD-5 were harvested at D8. All samples were collected in six biological replicates. Pelleted worms were lysed by beat beating in TRIzol reagent (Thermo Fisher, Waltham, MA). Total RNA was isolated using Nucleospin RNA (Macherey Nagel) following the manufacturer's protocol. Contaminant genomic DNA was removed from samples with TURBO DNase (Thermo Fisher) following the manufacturer's protocol. mRNA quality was assessed by NanoDrop 1000 v3.8 (Thermo Scientific), and 100 ng/µl of the samples with RNA integrity number above 8.0 were proceeded with library preparation. A paired-end bulk RNA sequencing for a minimum of 100 sequencing length per read was carried out using Illumina HISEQ 4000 sequences with 2 lanes per sample. Library preparation, RNA quality assessment, RNA-seq and quality control were outsourced from Genome Institute of Singapore, A*STAR (Singapore). Raw FASTQ reads and gene counts were uploaded to the NCBI GEO database with Accession number GSE182981.

The raw sequences for ND, HGD-1 and HGD-5 were aligned to WormBase reference genome (GCF_000002985.6) using *Rsubread* package (v2.4.3) and annotated with WormBase (WBcel235.104). Gene counts from two independent runs of bulk RNA sequencing generated from Binary Alignment/Map (BAM) files were normalised and con-founding factors such as batch variation were considered. The differential gene expression analysis using *DESeq2* (v1.30.1) during which undetected (NA) and lowly expressed genes (<10) were filtered out for at least five biological replicates.

An adjusted *P*-value of 0.05 was used to define differentially expressed genes, gene set enrichment analysis [Gene Ontology using *clusterProfiler* (v4.0.5); MeSH Terms using *meshes* (v1.18.1)] and pathway analysis [Kyoto Encyclopaedia of Genes and Genomes using *gage* (v2.42.0) and *pathview* (v1.32.0)]. All codes were written in RStudio (R version 4.0.4). Plots were generated using *pheatmap* (v1.0.12) and *ggplot2* (v3.3.5).

## Immunoblotting

Worms were collected, washed in M9 buffer, then lysed in RIPA buffer (50 mM Tris-HCl pH 7.5, 150 mM NaCl, 1% NP-40, 0.1% SDS, 2 mM EDTA, and 0.5% sodium deoxycholate) with protease inhibitor cocktail (Roche, Basel, Switzerland) by bead beating three times for 30 s at 6500 rpm with the samples chilled on ice between the homogenisation steps. Samples were then centrifuged 5 min at 10,000 × *g* 4 °C. Cleared lysate protein concentration was measured by using the BCA assay kit (Thermo Fisher, Waltham, MA). Sixty micrograms of total proteins were separated on 10% SDS-PAGE gels and transferred to nitrocellulose membranes. Membranes were blocked for 1 h with Odyssey blocking buffer TBS (Li-COR Biosciences, Lincoln, NE) at room temperature, and incubated with 1:1,000 of monoclonal anti-GFP antibody and 1:10,000 of monoclonal anti-tubulin antibody overnight at 4 °C (Roche, catalogue number 11814460001), washed, and incubated with 1:10,000 of IRDye 800CW anti-mouse IgG antibody (Li-COR Biosciences, catalogue number 925-32210). Membranes were washed and scanned with an Odyssey CLx imaging system. Uncropped and unprocessed scan of the cropped blot of Fig. 3b is reported in Supplementary Fig. 5.

## Trehalose assay

About 1000 worms were collected, washed in M9 buffer and lysed in RIPA buffer by bead beating three times for 30 s at 6500 rpm with the samples chilled on ice between the homogenisation steps. Samples were then centrifuged 5 min at 10,000 × *g*, 4 °C. *C. elegans* were harvested and washed off plates using M9 buffer. Cleared lysate protein concentration was measured using the BCA assay kit (Thermo Fisher, Waltham, MA). Remaining cleared lysates were treated with 10 mg/ml alkaline borohydrides to reduce sugar and subsequently neutralised with 200 mM acetic acid. Next, the concentrations of trehalose were measured using the Trehalose Assay Kit (Megazyme, Wicklow, Ireland) following the manufacturer's protocol using Tecan Infinite M200Pro with i-control software.

## Neutral lipid staining

Fat staining to visualise lipid droplet was performed as previously described[93]. Day 1 adult worms were collected and washed thrice with 1× PBS, allowed to settle by gravity, then permeabilized 1 h at room temperature with equal volume of 1× PBS and 2× MWRB buffer (160 mM KCl, 40 mM NaCl, 20 mM EDTA, 10 mM spermidine, 30 mM HEPES and 50% methanol) containing 2% paraformaldehyde. Permeabilized animals were washed thrice with 1× PBS. Three-hundred microliters of 60% isopropanol was then added to the worms and incubated for 15 min. Thereafter, palleted worms were incubated 2 h at room temperature in 1 mL Oil-Red-O, or ORO (Sigma, catalogue number O0625) working solution. The working solution was prepared by diluting 5 g/L ORO stock solution in isopropanol with water to a final concentration of 60%. Thereafter, the worms were allowed to settle with gravity, followed by at least two washes of 1× PBS + 0.01% Tween 20 and small number of worms were pipetted on 2% agarose pad on a glass slide for imaging. The samples were imaged with a Zeiss Cell Observer fluorescence microscope at 20X magnification (Zeiss) taken with a colour camera with same exposure settings used across all the conditions. In Fiji software version 1.53c, the RGB images were split into their respective channel while retaining the images from green channel for further analysis[94]. The images were then inverted and the intensity of the ORO staining immediately behind the pharynx of each worm was measured with the oval selection tool with fixed area of 70 pixels radius across all the samples. The intensity of ORO staining is derived from the mean intensity of ORO stain minus the average mean intensity of the background multiplied by the area of selection[55].

## Lipid extraction and total fatty acid analysis

Synchronised L1 worms were transferred on NGM plates seeded with UV-killed OP50 bacteria and grown to the L4 stage. Worms were then transferred to new plates containing 50 mM FUdR. On day 1 of adulthood, a subset of the worms was transferred to plates without glucose (ND-1) or with 2% glucose (HGD-1) for 24 h. Independently, on day 5 of adulthood, another subset of worms was transferred to plates without glucose (ND-5) and with 2% glucose (HGD-5) for 24 h. From these 4 conditions, approximately 10,000 worms were harvested, washed thoroughly with M9 buffer and homogenised by bead beating three times for 30 s at 6,500 rpm. A portion of the lysates was cleared to quantify total protein using the bicinchoninic acid (BCA) protein quantification assay (Sigma-Aldrich). A volume corresponding to three mg of total protein was transferred to glass tubes and lyophilised overnight (Virtis). One hundred microliters of 1 mM C23:0 (lignoceric acid) were added to the tubes as internal standard. The worms were hydrolysed and esterified to fatty acid methyl esters (FAME) with 500 µl of 1.25 M HCl-methanol for 1 h at 80 °C. FAMEs were extracted three times with 1 ml of hexane. Combined extracts were dried under nitrogen gas, resuspended in 15 µl hexane. FAMEs were separated by gas chromatography with a flame ionisation detector (GC-FID) (GC-2014; Shimadzu, Kyoto, Japan) using a ULBON HR-SS-10 50 m × 0.25 mm column (Shinwa, Tokyo, Japan). Supelco 37 component FAME mix was

used to identify corresponding fatty acids (Sigma-Aldrich, St. Louis, MO). Data were normalised to internal standards C15:0 and C23:0. Heatmap was generated in GraphPad Prism 9.2.0.

## Fluorescence microscopy

To quantify DAF-16::GFP localisation and GFP expression, worms were immobilised with 25 mM tetramisole and mounted on a 2% agarose pad. Images were captured using Zeiss LSM 800 confocal fluorescence microscope (Carl Zeiss AG, Oberkochen, Germany) with 20× HC PL objective lens with an excitation at 488 nm and emission at 550(13) nm. Two-micrometre Z-stack sections were merged and the number of cells with nuclear-localised DAF-16::GFP was quantified using Fiji software version 1.53c.

## Quantitative real-time PCR

Approximately 1,000 synchronised worms were harvested at the indicated time, and washed in M9 buffer. Worms were lysed with a motorised pestle homogeniser. Total RNA was isolated using TRIzol reagent (Thermo Fisher, Waltham, MA). Contaminant DNA was removed from samples with TURBO DNase (Thermo Fisher) following the manufacturer's protocol. Complementary DNA (cDNA) was synthesised from total RNA using RevertAid Reverse Transcriptase (Thermo Fisher, Waltham, MA) following the manufacturer's protocol. qPCR was performed with SYBR Green (Qiagen, CA, USA) according to the manufacturer's protocol using a CFX-96 Real-time PCR system (Bio-Rad, Hercules, CA, USA). Thirty nanograms of cDNA and 50 nM of the paired primer mix for target genes were used for each reaction. Relative mRNA was normalised to the housekeeping gene *act-1* (Supplementary Table 1). Oligonucleotides used in this study are listed in Table S2.

## Statistics

Error bars indicate standard error of the mean (SEM), calculated from at least three biological replicates unless otherwise indicated. *P* values were calculated using one-way ANOVA with Tukey's test or log-rank test for lifespan unless otherwise indicated and reported as *P* values with 4 significant digits in the figures. All statistical tests were performed using GraphPad Prism 9.2.0 software.

## Reporting summary

Further information on research design is available in the Nature Research Reporting Summary linked to this article.

# Data availability

The source data for this study are available at the Dataverse Project doi.org/10.21979/N9/GEPEAG and as a Source data file provided with this paper. All RNA sequencing data is available at the Gene Expression Omnibus accession number GSE182981. Source data are provided with this paper.

# Code availability

R codes used in this study for quantitative bulk RNA sequencing are available at http://github.com/cenk-celik/c_elegans_rnaseq.

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

## Acknowledgements

We are grateful to Drs. Sivan Henis-Korenblit and Seung-Jae V. Lee for helpful discussions and critical reading of the manuscript. We thank our colleagues Drs. Valerie Lin Chun Ling and I-Hsin Su as well as Thibault's lab members Xiu Hui Fun, Nurulain Ho and Peter Jr. Shyu for critical reading of the manuscript. We thank Dr. Irene Gallego Romero for helpful discussions on bulk RNA sequencing analysis. We also thank Dongyeop Lee and Dr. Seung-Jae V. Lee for generously sharing the glucose-feeding RNA sequencing data and glucose reporter strain *far-3p::GFP(bc14852)*[37]. We are thankful to Dr. Rebecca Taylor for sharing integrated *gly-19p::xbp-1s* and *rab-3p::xbp-1s* strains. Some strains were provided by the *Caenorhabditis* Genetics Centre, which is funded by NIH Office of Research Infrastructure Programs (P40 OD010440). This work was supported by the Singapore Ministry of Education Academic Research Fund Tier 2 (2018-T2-1-002) and Tier 1 (2019-T1-002-011) as well as the Ministry of Health, Singapore, National Medical Research Council Open Fund Individual Research Grant (MOH-000566).

## Author contributions

C.B.C., L.W. and G.T. designed the studies. C.B.C., L.W., A.A.K., S.T., and V.W.X.L. performed lifespan assays. C.B.C. prepared and isolated mRNA for bulk RNA sequencing. C.C. analysed data and generated figures from bulk RNA sequencing. L.W. performed qPCR experiments. C.B.C., L.W., J.H.K., A.D.L. carried out the biochemical and phenotype assays. S.X. prepared and acquired fluorescent images. All authors contributed to analysing the data. C.B.C, L.W. and G.T. wrote the original manuscript draft with input from all the authors. C.C. and G.T. wrote the revised manuscript.

## Competing interests

The authors declare no competing interests.
