## [Peer Review File · Nature Communications]

Reviewers' Comments:

Reviewer #1:

Remarks to the Author:
review

This manuscript is focused on *C. elegans* fed a bacterial diet with 2% glucose (high glucose diet, HGD.). The authors claim 7-day-old (7DO) post-reproductive animals show significant lifespan extension which is in contrast to a decreased lifespan of reproductive 3-day-old animals. Analysis with mutants suggest that *daf-2* and *age-1* but not *daf-16* are critical. Additional Studies identified the involvement of UPR activation including ATF-6 and PEK-1 . The study is interesting however there are serious Scientific flaws that makes the data/ conclusions difficult to assess and therefore the manuscript should be rejected.

1-the authors start the lifespan from the egg hatching event. This is very different than most lifespan analysis which begins post L4. Therefore, it is difficult to compare the days/conclusions with other studies. However, the authors use FUDR which prevents normal development of progeny. Therefore, how can one conclude anything about the reproductive period? This is a serious problem with all of the analysis! The life span analysis also does not have a wild type control.

2-the effects on the lifespan are very small. This reviewer is not convinced the data suggests the absence of the requirement for *daf-16* for example.

3-in the lifespan methods it states "Worms that were male or dead due to desiccation were excluded from the analysis". Why were there males?

4- the *daf-16* gfp photos need to be redone. The photos look like we are looking at dust specs not nuclei. They are not convincing at all.

5-*daf-16* is known to target hundreds of genes. Simply choosing two, to give you an assessment of gene function does not seem adequate.

6-Figure 2h seems disjointed from the text.

7- From the referenced paper A"CREBh is a Novel UPR Gene Dependent on *ire-1*, *xbp-1*, and *atf-6*- In the i-UPR gene list, we identified a gene—F57B10.1—encoding a bZIP transcription factor homologous to mammalian CREBh." Therefore, this is not an indicator of a single pathway.

8- did the authors compare the results from the screen with the results from the lab pf SJ Lee published a screen using *far-3*?

Reviewer #2:

Remarks to the Author:

In the manuscript, Lei et al. describe a very interesting phenomenon, where a high glucose diet (HGD), which historically has been represented to be toxic, can be beneficial at old age due. Perhaps most interesting, the authors find that this is independent of the IIS pathway and is actually due to the involvement of the UPRER. The study is extremely interesting, presents a highly novel finding, and provides a substantial contribution to the fields of IIS and UPRER. For these reasons, I strongly urge the editor to consider this manuscript for Nature Communications. However, the manuscript is missing critical controls that are essential to make the conclusions described. I believe that the recommendations outlined below will strengthen the manuscript, clarify the message, and make the manuscript applicable to a broader audience.

Major comments:

-The materials and methods states that FUDR was used to prevent progeny formation in worms. Considering that this manuscript argues that the effects of high-glucose on post-reproductive animals increases lifespans makes the use of FUDR for the lifespans in this manuscript

questionable, considering the fact that the younger animals are not necessarily reproductive if FUDR is used. Minimally, WT lifespan on ND vs. HGD at 7DO need to be redone without FUDR to make the argument that high-glucose is beneficial post-reproduction, or the manuscript needs to be rewritten to state that older animals only benefit from high-glucose (not post-reproductive).

- Moreover, if the argument of reproduction wants to be brought in, a better experiment would be to use mutants that fail at reproductive development completely, such as *glp-1* mutants. Since *glp-1* mutants already have a lifespan extension that is *daf-16* dependent, it would be interesting to see if high-glucose can have an additive effect on these animals. Moreover, if HGD can affect lifespan in germline-less mutants, does this only occur at old age or also at young age since they are non-reproductive, considering the arguments made in the manuscript?
- The manuscript states that the HGD-lifespan extension is *daf-16* independent, yet the lifespan extension of HGD on *daf-16* mutants is half that seen in WT conditions (6% versus 11% as per extended data table 1). This seems to imply that the lifespan extension is partially dependent on *daf-16*. The arguments made here would also be much clearer if a WT control were actually placed here.
- The first paragraph on page 7 describing figures 2d-g are very confusing, and the way it is written, it sounds contradictory. The authors state that for some things (such as *daf-16* nuclear localization and *sip-1* expression), *daf-16* is not active in 7DO worms, but for other things (*sod-3* expression), *daf-16* is active. They then make the conclusion that the IIS pathway remains intact and functional in aged worms. Finally, the last panel of pulling in tunicamycin-induced *skn-1* does not seem to fit in here. The authors need to make some effort to clean up this section and present the data in a clearer manner either textually or through new experiments (look at a larger panel of *daf-16* targets or some other clearer way to represent that *daf-16* is actually active in 7DO animals – I am not convinced the way the data is presented).
- A similar situation to *daf-16* mutants in Fig. 2C occurs in Figure 3C where *ire-1* mutants have a partial phenotype as well (5% versus 11% in WT). Perhaps it may be beneficial to look at the *xbp-1* mutant as well for a clearer result (since *xbp-1* is more specific to UPRER activation, while *ire-1* has other functions, such as RIDD). Again, having a WT control is essential. Moreover, in the screen presented in Fig. 4, IRE-1 seems to play a pivotal role in activation of the *far-3p::GFP* glucose reporter, pushing even further the possibility that IRE-1 may play at least a partial role in the model presented in this manuscript.
- The model argues that HGD at young age would result in chronic ER stress, which is toxic, while at late age, it increases UPR to promote longevity. This model would be strengthened if a model for constitutive UPRER is tested. Would an animal with constitutive UPRER be resistant to HGD-induced ER toxicity at young age? And would an animal with constitutive UPRER have no change from HGD at late age?

Minor comments:

- Last sentence in introduction (pg. 4) – did you mean to say “HGD was not dependent on *daf-16* (FOXO)?
- I am assuming that the *daf-2(e1370)* strain is used in this manuscript as that is the strain listed under the materials and methods section. This is one of the longest-lived mutants that exists, yet in Extended Data Fig. 2, the *daf-2(lof)* and WT on ND has the same lifespan. Can you comment on this? Also, *age-1* mutants exhibit extended lifespans in previous reports, yet a comparison of Figure 1a and Figure 2b suggests that you do not see a lifespan extension in *age-1* mutants. Please comment on this. Once again, a WT control would provide clarity here.

Reviewer #3:

Remarks to the Author:

In the manuscript entitled “Glucose increases the lifespan of post-reproductive *C. elegans* independently of FOXO” authors provides evidence sustaining that a high glucose diet in aged nematodes can extend lifespan as opposed to young animals, where such regimen is a well-established inducer of lifespan shortening. Authors attempt to dissect the molecular bases of such findings by tacking the classical *daf-2/ daf-16* pathway, which was also shown to mediate lifespan

regulation in previous publications from different groups. Additionally, authors aim to analyze the contribution of UPR sensors in high glucose diet mediated lifespan extension in aged organisms showing that ablating *atf-6* and *PEK* blocks high glucose diet effects on lifespan in aged nematodes, which was not observed in *IRE1* mutants. Authors perform RNAi screening followed by gene ontology and tissue enrichment analysis to understand how UPR mediates such effects and conclude that UPR plays a role in maintaining ER homeostasis in the intestine.

Overall, results are properly presented; however, for the standards of Nature Communication, authors were expected to evaluate deeply the molecular mechanisms behind lifespan extension in post-reproductive nematodes following HFD as this interesting result is of great interest to general scientific community studying metabolic alterations during aging. At least, a transcriptomic and/or proteomic analysis should be performed in different nematodes tissue (the intestine and brain, for instance) to further understand what are the molecular basis of this observation. The contribution of UPR mediators to the phenomenon were not fully covered as conditional mutants of UPR mediators with expression in the intestine or brain tissue would clarify how ER stress response is transmitted from different tissues as described in the seminal 2013 paper of Rebeca Taylor and Dillin (doi:10.1016/j.cell.2013.05.042 (2013)).

Despite the interesting findings regarding lifespan extension in high glucose input in aged worms, I would not recommend this manuscript for publication in Nature Communications.

Minor comments:

1. Did authors measure levels of *daf-2* and *age-1* following HGD? Do they go to the nucleus? Authors performed this assay for *daf-16* but not for *daf-2* and *age-1*. What happens with *daf-16* mutants if submitted to the same unconventional temperature shift implemented to *daf-2* evaluation?
2. Authors should confirm the finding that *ATF-6* and *PEK* activation compensate for weak activation of *IRE1* by treating worms with tunicamycin. Would *IRE1* show an increased disruption compared to *atf6* and *pek* activation? if it does not, maybe the findings in HGD regarding UPR activation are specific to *IRE1* pathway in the context of glucose metabolism.
3. It is interesting to note that authors sustain that *IRE1* doesn't have a role in extending lifespan following HGD, however, Taylor paper shows that *XBP1s* is necessary to increase lifespan by cell non autonomous responses in the intestine in a normal context. How do authors interpret this data?

REMARKS TO AUTORS

Reviewer 1 comments

This manuscript is focused on *C. elegans* fed a bacterial diet with 2% glucose (high glucose diet, HGD.). The authors claim 7-day-old (7DO) post-reproductive animals show significant lifespan extension which is in contrast to a decreased lifespan of reproductive 3-day-old animals. Analysis with mutants suggest that *daf-2* and *age-1* but not *daf-16* are critical. Additional Studies identified the involvement of UPR activation including ATF-6 and PEK-1. The study is interesting however there are serious scientific flaws that makes the data/ conclusions difficult to assess and therefore the manuscript should be rejected.

1. the authors start the lifespan from the egg hatching event. This is very different than most lifespan analysis which begins post L4. Therefore, it is difficult to compare the days/conclusions with other studies. However, the authors use FUDR which prevents normal development of progeny. Therefore, how can one conclude anything about the reproductive period? This is a serious problem with all of the analysis! The life span analysis also does not have a wild type control.

We agree with the reviewer that reporting lifespan from L1 was confusing. We have modified the manuscript and the figures accordingly starting lifespan analysis from young adults. In the revised manuscript, we have included 8 biological replicates of wild-type (WT) animal fed normal diet (ND) and high glucose diet from day 5 of adulthood (HGD-5) (ND, n=722; HGD-5, n=772 including biological replicates). See Reviewer #1, comment #2 for details on reproducibility. Also, we would like to emphasize that animals were exposed to FUDR (when indicated) only from L4 stage in all lifespans.

We also agree with the reviewer that our original manuscript lacked the experiments supporting our claims of extended lifespan in post-reproductive animals subjected to HGD. In this revised manuscript, we described these animals as aged animals (HGD-5). In addition, we have carried out lifespan assays in WT (Fig. 4a, Supplementary Fig. 3a and Table 1), germline defective *glp-1(lof)* (Fig. 4b and Supplementary Table 1), and WT male (Fig. 4c, Supplementary Fig. 3b and Table 1) animals in the absence of 5-fluoro-2'-deoxyuridine (FUDR) as well as in WT animals in the presence of FUDR only for the first 8 days of adulthood (Fig. 4d and Supplementary Table 1). Here is a summary of our findings:

- The lifespan of WT hermaphrodite animals on HGD-5 was dramatically reduced in the absence of FUDR and was similar to the lifespan of animals on HGD-1 (Fig. 4a, Supplementary Fig. 3a and Table 1).
- The lifespan of *glp-1(lof)* animals fed HGD-5 was significantly shorter compared to HGD-1 in the absence of FUDR (Fig. 4b and Supplementary Table 1).
- The lifespans of WT males fed HGD-1 and HGD-5 were extended compared to ND (Fig. 4c, Supplementary Fig. 3b and Table 1).
- Aged WT hermaphrodite animals on HGD (HGD-5) lived longer compared to ND when exposed to FUDR from L4 to days 7 of adulthood (Fig. 4b and Supplementary Table 1).

Updated manuscript (lines 188-219)

*“Five-fluoro-2'-deoxyuridine (FUDR) is a commonly used reagent for chemically sterilising adult worms in lifespan assays³⁹. However, FUDR alone has been shown to extend lifespan by promoting stress responses in mutant animals or during stressing growth conditions^{40, 41, 42, 43, 44}. Therefore, we asked if there is a synergic effect between FUDR and HGD in promoting the longevity of HGD-5 animals. In the absence of FUDR, the lifespan of WT animals on ND was comparable to those grown in the presence of FUDR (Fig. 4b, Supplementary Table 1), in agreement with previous reports^{42, 43}. Similarly, animals on HGD-1 exhibited a shorter lifespan compared to ND. Conversely, the lifespan of HGD-5 animals was drastically shortened in the absence of FUDR and comparable to the lifespan of animals on HGD-1. To further assess the role of fertility in modulating lifespan, we used long-lived germline defective *glp-1(loss-of-function; lof)* animals⁴⁵. Surprisingly, the lifespan of *glp-1(lof)* animals fed on HGD by day 5 was significantly shorter compared to HGD-1 in the absence of FUDR (Fig. 4c, Supplementary Table 1). Together, these findings suggest that germline proliferation plays a role in promoting the longevity of HGD-5 animals.*

To further dissect HGD-induced longevity to the role of the germline in modulating lifespan, we performed lifespan assays in male worms. The lifespan of male WT animals fed ND was mildly reduced by FUDR (Supplementary Fig. 3a and Table 1). Next, we monitored the lifespan of WT males fed HGD in the absence of FUDR (Fig. 4d, Supplementary Table 1). As previously reported⁴⁶, the lifespan of male WT worms fed on HGD-1 was extended compared to ND while remaining comparable to the lifespan of HGD-5. In the presence of FUDR, lifespans of either HGD-1 or HGD-5 males were still extended compared to ND (Supplementary Fig. 3b and Table

1). Together, these findings suggest that HGD promote longevity in young and aged male worms independently of FUDR.

By convention, FUDR is introduced to the animals from larva stage 4 (L4) and supplemented throughout adulthood^{47, 48}. However, *C. elegans* lays eggs about the first four days of adulthood⁴⁹. Therefore, we interrogated if FUDR affects the longevity of animals beyond the fertility window. WT animals fed on ND, HGD-1 or HGD-5 were exposed to FUDR from L4 to days 7 of adulthood (Fig. 4e, Supplementary Table 1). The lifespan of HGD-1 animals was slightly shorter than ND, while HGD-5 animals lived longer compared to ND. Interestingly, the total brood size of HGD-1 animals was dramatically reduced by FUDR while it remained unchanged in ND animals in the absence and presence of FUDR (Supplementary Fig. 3c). Together with the lifespan assays in the absence or presence of FUDR, findings suggest that embryogenesis counteracts life extension induced by HGD in aged animals."

New Figure 4b-e:

New Supplementary Figure 3a-b:

2. the effects on the lifespan are very small. This reviewer is not convinced the data suggests the absence of the requirement for *daf-16* for example.

We understand the reviewer concerned that the effects on the lifespan might appear small in some of reported assays. Initially, we had the same concerns. We accidentally observed that aged animals were living longer on HGD compared to ND as it was intended for a different project. To validate our initial observation, the lifespan assays of aged WT fed ND or HGD has been repeated independently by 3 lab members with similar outcomes from 8 biological replicates (Rebuttal Table 1; ND, n=722; HGD-5, n=772 including biological replicates). Together, these assays robustly support our conclusion that aged WT animals have an extended lifespan on HGD compared to ND.

Rebuttal Table 1. Lifespan analysis.

Figure	Strains	Drug	Treatment*	Mean lifespan \pm SEM (days)	75%	% change to control	Number of animals	P values versus ND	Biological replicates
1b	WT	FUdR	ND	21.3 \pm 0.5	25	-	113/120	-	Fig. 1b
			HGD-5	23.6 \pm 0.4	26	10.7	116/120	0.0065	
	WT	FUdR	ND	19.8 \pm 1.5	25	-	45/80	-	Fig. 1b bio. rep. 2
			HGD-5	23.7 \pm 1.4	31	20	70/100	0.0007	
	WT	FUdR	ND	20.2 \pm 0.5	24	-	67/85	-	Fig. 1b bio. rep. 3
			HGD-5	22.4 \pm 0.9	29	11	50/92	0.0016	
	WT	FUdR	ND	22.4 \pm 0.5	26	-	80/84	-	Fig. 1b bio. rep. 4
			HGD-5	25.0 \pm 0.9	28	12.3	77/84	0.0085	
	WT	FUdR	ND	18.5 \pm 0.3	19	-	79/120	-	Fig. 1a,b bio. rep. 5
			HGD-5	20.1 \pm 0.4	22	8.1	79/120	0.0014	
	WT	FUdR	ND (L)	29.5 \pm 1.5	41	-	61/61	-	Fig. 1b bio. rep. 6
			HGD-5 (L)	38.0 \pm 1.3	47	28.8	82/82	<0.0001	
	WT	FUdR	ND (L)	31.3 \pm 0.7	38	-	117/117	-	Fig. 1b bio. rep. 7
			HGD-5 (L)	35.7 \pm 0.9	41	14.3	128/128	<0.0001	
	WT	FUdR	ND (L)	31.2 \pm 1.1	41	-	121/121	-	Fig. 1b bio. rep. 8
			HGD-5 (L)	35.6 \pm 1.0	45	13.8	132/132	0.0094	

*ND, normal diet; HGD, high glucose diet; HGD-1, HGD-5, treatment at day 1, or 5 adult worms, respectively; L, indicates lifespan assays performed in liquid culture.

Similar to WT, we performed the lifespan assays of aged *daf-16(1of)* fed ND and HGD 4 times (total of > 350 animals). The lifespan of *daf-16(1of)* animals was significantly extended when fed HGD compared to ND (Rebuttal Table 2; ND, n=502; HGD-5, n=536 including 5 biological replicates). To complement these findings, we quantified the localisation of DAF-16::GFP. We reasoned that DAF-16::GFP shouldn't localise to the nucleus in aged animal fed HGD as *daf-16* is dispensable to extend the lifespan of HGD-5 animals.

Rebuttal Table 2. Lifespan analysis.

Figure	Strains	Drug	Treatment*	Mean lifespan \pm SEM (days)	75%	% change to control	Number of animals	P values versus ND	Biological replicates
5e	daf-16(1of)	FUdR	ND	12.7 \pm 0.2	13	-	120/120	-	Fig. 5e
			HGD-5	15.0 \pm 0.3	18	16	116/120	<0.0001	
	daf-16(1of)	FUdR	ND	16.2 \pm 0.5	20	-	38/77	-	Fig. 5e bio. rep. 2
			HGD-5	18.0 \pm 0.9	22	11.3	47/107	0.0356	
	daf-16(1of)	FUdR	ND	16.7 \pm 0.2	19	-	124/137	-	Fig. 5e bio. rep. 3
			HGD-5	17.7 \pm 0.3	19	6	154/172	0.0018	
	daf-16(1of)	FUdR	ND	14.8 \pm 0.2	15	-	120/120	-	Fig. 5e bio. rep. 4
			HGD-5	17.0 \pm 0.2	19	15.2	120/120	<0.0001	
	daf-16(1of)	FUdR	ND	19.8 \pm 0.3	21	-	100/100	-	Fig. 5e bio. rep. 5
			HGD-5	21.9 \pm 0.3	23	10.4	99/100	<0.0001	

*ND, normal diet; HGD, high glucose diet; HGD-1, HGD-5, treatment at day 1, or 5 adult worms, respectively.

Additionally, DAF-16 was not activated in HGD-5 as someone would have expected if DAF-16 plays a role in extending the lifespan of HGD-5. In contrast, DAF-16 was inhibited upon HGD compared to ND in 5 day adulthood (Fig.5 f-h). As expected, DAF-16::GFP strongly localised to the nucleus in HGD-1 and HGD-5 worms upon heat shock, indicating DAF-16 activation is not compromised with age.

New Figure 5f-h:

Likewise, our RNA-seq data analysis suggest that the DAF-16 pathway is activated mostly in HGD-1 (Fig. 5a-d). Overall, more genes were upregulated by DAF-16 in HGD-1 than HGD-5, suggesting that DAF-16 might not modulate HGD-5 lifespan. Similarly, the insulin/insulin-like growth factor (IGF-1) signalling (IIS) pathway as well as heat shock factor downstream target genes (HSF-1) were differentially regulated in HGD-1 compared to ND and HGD-5. Together, these findings suggest that DAF-16 is excluded from extending the lifespan of D5 animals fed HGD.

New Figure 5a-d:

3. in the lifespan methods it states "Worms that were male or dead due to desiccation were excluded from the analysis". Why were there males?

We thank the reviewer for pointing this out. In the original submission, we reported the lifespan from L1. The proportion of male *C. elegans* is usually low under normal conditions (0.01-0.1%). Therefore, someone cannot exclude the possibility of finding males in the assays. In our hands, we have not seen any males when carrying out lifespan assays of *C. elegans* hermaphrodites. We included "males" in the original method as cited by other research groups. In the revised manuscript, we have amended the "Lifespan assay on solid plates" in the method section as follow:

"Worms that were dead due to desiccation were excluded from the analysis" (lines 740-741)

4. the *daf-16* gfp photos need to be redone. The photos look like we are looking at dust specs not nuclei. They are not convincing at all.

We agree with the reviewer that DAF-16::GFP signals were below standard. The assay has been repeated in biological triplicates in this revised manuscript (Fig. 5f-h) with similar results. We included WT fed NO at 1 or 5 days of adulthood (D1 and D5 respectively) to indicate the autofluorescence of young and aged animals (Fig. 5f, white dashed lines).

To quantify the activation of DAF-16 in D5 worms, we visualised DAF-16::GFP localisation to the nucleus (Fig. 5f-h). In D1 worms fed 16h HGD, we observed a significant increase in nuclear DAF-16::GFP localisation, indicating DAF-16 activation by HGD. In contrast, the number of DAF-16::GFP nuclear localised cells in HGD-5 was significantly lower compared to ND. As expected, DAF-16::GFP strongly

localised to the nucleus in HGD-1 and HGD-5 worms upon heat shock, indicating DAF-16 activation is not compromised with age. Together, these findings suggest that the DAF-2/DAF-16 axis is not the main driver to extend the lifespan of D5 animals on HGD.

New Figure 5f-h:

5. *daf-16* is known to target hundreds of genes. Simply choosing two, to give you an assessment of gene function does not seem adequate.

We agree with the reviewer that we oversimplified the role of *daf-16* in our original submission. We removed the data presented in Fig. 2f-h of our original submission. To address this issue, we carried out quantitative transcriptomic in WT animals (Fig. 2a,b). A single pool of synchronised worms was grown on ND until day 1 of adulthood (D1). From the same single pool, a subset of worms was fed on HGD at day 1 (HGD-1). The second subset of worms grown on ND was fed on HGD at day 5 (HGD-5) of adulthood (D5). Worms fed ND, HGD-1, and HGD-5 were harvested at D8 (n=6).

New Figure 2a,b:

Our RNA-seq data analysis suggest that the DAF-16 pathway is activated mostly in HGD-1 (Fig. 5a,b). Overall, more genes were upregulated by DAF-16 in HGD-1 than HGD-5, suggesting that DAF-16 might not modulate HGD-5 lifespan. Similarly, the insulin/insulin-like growth factor (IGF-1) signalling (IIS) pathway was differentially regulated in HGD-1 compared to ND and HGD-5. Together with the lifespan and DAF-16 localisation, it suggests that DAF-16 is excluded from extending the lifespan of D5 animals fed HGD.

New Figure 5a,b:

6. Figure 2h seems disjointed from the text.

As mentioned in our response to comment #5 of Reviewer #1, we removed the data that was presented in Fig. 2f-h from our original submission.

7. From the referenced paper "CREBh is a Novel UPR Gene Dependent on ire-1, xbp-1, and atf-6- In the i-UPR gene list, we identified a gene—F57B10.1—encoding a bZIP transcription factor homologous to mammalian CREBh." Therefore, this is not an indicator of a single pathway.

Indeed, we originally tested several possible target genes of ATF-6 and PEK-1 branches of the unfolded protein response (UPR) based on published data (PMID 16184190). To validate that the genes *cht-1* and *cbp-3* are specifically upregulated by ATF-6 and PEK-1, respectively, we induced the UPR with the drug tunicamycin (Tm) in *atf-6(lox)* and *pek-1(lox)*. We observed no significant induction in the expression of *cht-1* and *cbp-3* in *atf-6(lox)* and *pek-1(lox)* in animals subjected to Tm compared to carrier (Supplementary Fig. 4a-c). These findings indicate that both target genes are specifically upregulated by the UPR. However, it does not exclude that these genes can be upregulated by other pathways as it is the case for the well established UPR target gene *hsp-4*. The gene *hsp-4* is upregulated by IRE-1 upon endoplasmic reticulum (ER) stress as well as the heat shock response.

Supplementary Figure 4a-c:

8. did the authors compare the results from the screen with the results from the lab pf SJ Lee published a screen using *far-3*?

Indeed, we based our RNAi screen on the findings of Seung-Jae Lee group (PMID 26637528). Prof. Lee generously shared his strain *far-3p::GFP* with us. We shortlisted genes candidates by comparing genes that were commonly identified from their *far-3p::GFP* screen and from UPR-regulated genes from Randal Kaufman group (PMID 16184190) and from our previous publication (PMID 30333136). We identified 220 and 186 genes that were upregulated and downregulated, respectively, by the UPR while modulating the levels of the reporter GFP driven by the promoter *far-3*. In this revised manuscript, we decided to remove the RNAi screen as it does not fit very well with the rest of the data and we feel that it is too distracting. However, we used *far-3p::GFP* strain to validate our RNA seq data as well as to show that *far-3* is induced in a similar fashion in HGD-1 and HGD-5 animals

New Figure 3b:

b

Reviewer 2 comments

In the manuscript, Lei et al. describe a very interesting phenomenon, where a high glucose diet (HGD), which historically has been represented to be toxic, can be beneficial at old age due. Perhaps most interesting, the authors find that this is independent of the IIS pathway and is actually due to the involvement of the UPRER. The study is extremely interesting, presents a highly novel finding, and provides a substantial contribution to the fields of IIS and UPRER. For these reasons, I strongly urge the editor to consider this manuscript for Nature Communications. However, the manuscript is missing critical controls that are essential to make the conclusions described. I believe that the recommendations outlined below will strengthen the manuscript, clarify the message, and make the manuscript applicable to a broader audience.

Major comments:

1. The materials and methods states that FUDR was used to prevent progeny formation in worms. Considering that this manuscript argues that the effects of high-glucose on post-reproductive animals increases lifespans makes the use of FUDR for the lifespans in this manuscript questionable, considering the fact that the younger animals are not necessarily reproductive if FUDR is used. Minimally, WT lifespan on ND vs. HGD at 7DO need to be redone without FUDR to make the argument that high-glucose is beneficial post-reproduction, or the manuscript needs to be rewritten to state that older animals only benefit from high-glucose (not post-reproductive).

We agree with the reviewer that our original manuscript lacked the experiments supporting our claims of extended lifespan in post-reproductive animals subjected to HGD. In this revised manuscript, we described these animals as aged animals (HGD-5). In addition, we have carried out lifespan assays in WT (Fig. 4a, Supplementary Fig. 3a and Table 1), germline defective *glp-1(lof)* (Fig. 4b and Supplementary Table 1), and WT male (Fig. 4c, Supplementary Fig. 3b and Table 1) animals in the absence of 5-fluoro-2'-deoxyuridine (FUdR) as well as in WT animals in the presence of FUdR only for the first 8 days of adulthood (Fig. 4d and Supplementary Table 1). Here is a summary of our findings:

- The lifespan of WT hermaphrodite animals on HGD-5 was dramatically reduced in the absence of FUdR and was similar to the lifespan of animals on HGD-1 (Fig. 4a, Supplementary Fig. 3a and Table 1).
- The lifespan of *glp-1(lof)* animals fed HGD-5 was significantly shorter compared to HGD-1 in the absence of FUdR (Fig. 4b and Supplementary Table 1).
- The lifespans of WT males fed HGD-1 and HGD-5 were extended compared to ND (Fig. 4c, Supplementary Fig. 3b and Table 1).
- Aged WT hermaphrodite animals on HGD (HGD-5) lived longer compared to ND when exposed to FUdR from L4 to days 7 of adulthood (Fig. 4b and Supplementary Table 1).

Updated manuscript (lines 188-219)

*“Five-fluoro-2'-deoxyuridine (FUdR) is a commonly used reagent for chemically sterilising adult worms in lifespan assays³⁹. However, FUdR alone has been shown to extend lifespan by promoting stress responses in mutant animals or during stressing growth conditions^{40, 41, 42, 43, 44}. Therefore, we asked if there is a synergic effect between FUdR and HGD in promoting the longevity of HGD-5 animals. In the absence of FUdR, the lifespan of WT animals on ND was comparable to those grown in the presence of FUdR (Fig. 4b, Supplementary Table 1), in agreement with previous reports^{42, 43}. Similarly, animals on HGD-1 exhibited a shorter lifespan compared to ND. Conversely, the lifespan of HGD-5 animals was drastically shortened in the absence of FUdR and comparable to the lifespan of animals on HGD-1. To further assess the role of fertility in modulating lifespan, we used long-lived germline defective *glp-1(loss-of-function; lof)* animals⁴⁵. Surprisingly, the lifespan of *glp-1(lof)* animals fed on HGD by day 5 was significantly shorter compared to HGD-1 in the absence of FUdR (Fig. 4c, Supplementary Table 1). Together, these findings suggest that germline proliferation plays a role in promoting the longevity of HGD-5 animals.*

To further dissect HGD-induced longevity to the role of the germline in modulating lifespan, we performed lifespan assays in male worms. The lifespan of male WT animals fed ND was mildly reduced by FUdR (Supplementary Fig. 3a and Table 1). Next, we monitored the lifespan of WT males fed HGD in the absence of FUdR (Fig. 4d, Supplementary Table 1). As previously reported⁴⁶, the lifespan of male WT worms fed on HGD-1 was extended compared to ND while remaining comparable to the lifespan of HGD-5. In the presence of FUdR, lifespans of either HGD-1 or HGD-5 males were still extended compared to ND (Supplementary Fig. 3b and Table 1). Together, these findings suggest that HGD promote longevity in young and aged male worms independently of FUdR.

By convention, FUDR is introduced to the animals from larva stage 4 (L4) and supplemented throughout adulthood^{47, 48}. However, *C. elegans* lays eggs about the first four days of adulthood⁴⁹. Therefore, we interrogated if FUDR affects the longevity of animals beyond the fertility window. WT animals fed on ND, HGD-1 or HGD-5 were exposed to FUDR from L4 to days 7 of adulthood (Fig. 4e, Supplementary Table 1). The lifespan of HGD-1 animals was slightly shorter than ND, while HGD-5 animals lived longer compared to ND. Interestingly, the total brood size of HGD-1 animals was dramatically reduced by FUDR while it remained unchanged in ND animals in the absence and presence of FUDR (Supplementary Fig. 3c). Together with the lifespan assays in the absence or presence of FUDR, findings suggest that embryogenesis counteracts life extension induced by HGD in aged animals."

New Figure 4b-e:

New Supplementary Figure 3a-b:

2. Moreover, if the argument of reproduction wants to be brought in, a better experiment would be to use mutants that fail at reproductive development completely, such as *glp-1* mutants. Since *glp-1* mutants already have a lifespan extension that is *daf-16* dependent, it would be interesting to see if high-glucose can have an additive effect on these animals. Moreover, if HGD can affect lifespan in germline-less mutants, does this only occur at old age or also at young age since they are non-reproductive, considering the arguments made in the manuscript?

We have addressed the reviewer suggestion in the comment #1 above.

3. The manuscript states that the HGD-lifespan extension is *daf-16* independent, yet the lifespan extension of HGD on *daf-16* mutants is half that seen in WT conditions (6% versus 11% as per extended data table 1). This seems to imply that the lifespan extension is partially dependent on *daf-16*. The arguments made here would also be much clearer if a WT control were actually placed here.

We agree with the reviewer that our original data did not meet the expected standards. In this revised manuscript, we have included 8 biological replicates (ND, n=722; HGD-5, n=772 including biological replicates) of wild-type (WT) animal fed normal diet (ND) and high glucose diet from day 5 of adulthood (HGD-5) (Rebuttal Table 1; ND, n=722; HGD-5, n=772 including 8 biological replicates). We also have included 5 biological replicates (total of > 495 animals) of *daf-16(lof)* fed ND and HGD-5 (Rebuttal Table 2; ND, n=502; HGD-5, n=536 including 5 biological replicates). The percentage change of HGD-5 to ND in the lifespan of WT varies between 10% and 20% with a median of 11%. The percentage change of HGD-5 to ND in the lifespan of *daf-16(lof)* varies between 6% and 14% with a median of 10%. Overall, the percentage changes in *daf-16(lof)* compared to WT are non-significant (P value of 0.3595 with two-tailed distributions and two samples with equal variance). We have performed the lifespan assay of WT with *daf-16(lof)* of the replicates shown in Rebuttal Table 3.

Additionally, we quantified DAF-16::GFP localisation to the nucleus (Fig. 2c,d). In D1 worms fed 16h HGD, we observed a significant increase in nuclear DAF-16::GFP localisation, indicating DAF-16 activation by HGD. In contrast, the number of DAF-16::GFP nuclear localised cells in HGD-5 was significantly lower compared to ND. As expected, DAF-16::GFP strongly localised to the nucleus in HGD-1 and HGD-5 worms upon heat shock, indicating DAF-16 activation is not compromised with age.

Together, these findings suggest that the DAF-2/DAF-16 axis is not the main driver to extend the lifespan of D5 animals on HGD.

Rebuttal Table 3. Lifespan analysis.

Figure	Strains	Drug	Treatment*	Mean lifespan \pm SEM (days)	75%	% change to control	Number of animals	P values versus ND	Biological replicates
1b	WT	FUdR	ND	20.2 \pm 0.5	24	-	67/85	-	Fig. 1b bio. rep. 3
			HGD-5	22.4 \pm 0.9	29	11	50/92	0.0016	
5e	daf-16(1of)	FUdR	ND	16.2 \pm 0.5	20	-	38/77	-	Fig. 5e bio. rep. 2
			HGD-5	18.0 \pm 0.9	22	11.3	47/107	0.0356	

*ND, normal diet; HGD, high glucose diet; HGD-1, HGD-5, treatment at day 1, or 5 adult worms, respectively.

4. The first paragraph on page 7 describing figures 2d-g are very confusing, and the way it is written, it sounds contradictory. The authors state that for some things (such as *daf-16* nuclear localization and *sip-1* expression), *daf-16* is not active in 7DO worms, but for other things (*sod-3* expression), *daf-16* is active. They then make the conclusion that the IIS pathway remains intact and functional in aged worms. Finally, the last panel of pulling in tunicamycin-induced *skn-1* does not seem to fit in here. The authors need to make some effort to clean up this section and present the data in a clearer manner either textually or through new experiments (look at a larger panel of *daf-16* targets or some other clearer way to represent that *daf-16* is actually active in 7DO animals – I am not convinced the way the data is presented).

We thank the reviewer for pointing this out. Together with reviewer #1 comments #5 and #6, we removed the data presented in Fig. 2f-h from our original submission. To address this issue, we carried out quantitative transcriptomic in WT animals (Fig. 2a,b). A single pool of synchronized worms was grown on ND until day 1 of adulthood (D1). From the same single pool, a subset of worms was fed on HGD at day 1 (HGD-1). The second subset of worms grown on ND was fed on HGD at day 5 (HGD-5) of adulthood (D5). Worms fed ND, HGD-1, and HGD-5 were harvested at D8 (n=6).

New Figure 2a,b:

Our RNA-seq data analysis suggest that the DAF-16 pathway is activated mostly in HGD-1 (Fig. 5a,b). Overall, more genes were upregulated by DAF-16 in HGD-1 than HGD-5, suggesting that DAF-16 might not modulate HGD-5 lifespan. Similarly, the insulin/insulin-like growth factor (IGF-1) signalling (IIS) pathway was differentially regulated in HGD-1 compared to ND and HGD-5. Together with the lifespan and DAF-16 localisation, it suggests that DAF-16 is excluded from extending the lifespan of D5 animals fed HGD.

New Figure 5a,b:

5. A similar situation to *daf-16* mutants in Fig. 2C occurs in Figure 3C where *ire-1* mutants have a partial phenotype as well (5% versus 11% in WT). Perhaps it may be beneficial to look at the *xbp-1* mutant as well for a clearer result (since *xbp-1* is more specific to UPRER activation, while *ire-1* has other functions, such as RIDD). Again, having a WT control is essential. Moreover, in the screen presented in Fig. 4, IRE-1 seems to play a pivotal role in activation of the *far-3p::GFP* glucose reporter, pushing even further the possibility that IRE-1 may play at least a partial role in the model presented in this manuscript.

We thank the reviewer for these critical comments. In the original manuscript, we only compared the lifespans of *atf-6(lox)*, *ire-1(lox)*, and *pek-1(lox)* on ND and HGD-5. In the revised manuscript, we performed several additional lifespans of mutant animals related to the UPR. First, we extended the lifespans of *atf-6(lox)*, *ire-1(lox)*, and *pek-1(lox)* to HGD-1 (Figure 6d-f). Surprisingly, the lifespan of *ire-1(lox)* animals subjected to HGD-1 was significantly longer than animals subjected to ND and HGD-5 (Figure 6f). As suggested by the reviewer, we further characterise the role of IRE-1 pathway in modulating the longevity of animals subjected to HGD. The lifespan of *xbp-1(lox)* animals was shorter compared to WT on ND (Fig. 6g). The lifespan of *gly-19p::xbp-1s* (constitutive expression of spliced *xbp-1* in intestinal cells) animals was shorter while *rab-3p::xbp-1s* (constitutive expression of spliced *xbp-1* in neuronal cells) animals exhibited a significant increase in longevity when compared to WT on ND. Finally, the lifespan of *jnk-1(lox)* animals on ND was slightly shorter compared to WT. The lifespan of *xbp-1(lox)* animals was exacerbated upon HGD-1 when compared to WT animals on HGD-1 while the difference in lifespans were minor between both strains on HGD-5 (Fig. 6h,i). This suggests that XBP-1 is necessary to extend the lifespan of aged animals on HGD while IRE-1 might hyperactivate the UPR through *xbp-1s* in certain tissues. In agreement with this hypothesis, the constitutive expression of *xbp-1s* in neuronal cells increased the lifespan of *rab-3p::xbp-1s* animals on HGD-1 and to a less extent on HGD-5 compared to ND. On the other hand, the constitutive expression of *xbp-1s* in intestinal cells had little effect on the lifespans of *gly-19p::xbp-1s* animals on HGD when compared to WT animals. Similarly, *jnk-1(lox)* animals exhibit similar lifespans on the different diets in comparison to WT animals. Conjointly with the transcriptomic analysis, these findings suggest that IRE-1 exacerbates the lifespan of HGD-1 animals, while ATF-6 and PEK-1 promote longevity of HGD-5 animals.

New Figure 6d-i:

Updated manuscript (lines 268-298)

“Hereafter, we asked if the UPR renders a role in extending the lifespan of HGD-5 animals. The lifespans of *atf-6(lox)* and *pek-1(lox)* animals on HGD from D1 or D5 were similar to ND (Fig. 6d,e, Supplementary Table 1). These findings indicate that both ATF-6 and PEK-1 participate in extending the longevity of D5 worms on HGD. In contrast, HGD extended the lifespan of D5 *ire-1(lox)* animals when compared to ND (Fig. 6f, Supplementary Table 1). Surprisingly, the lifespan

of *ire-1(lof)* animals on HGD-1 was longer than *ire-1(lof)* animals on ND and HGD-5 as well as WT animals on ND (Supplementary Fig. 4d and Table 1).

To further characterise the role of IRE-1 pathway in modulating the longevity of animals subjected to HGD, we conducted lifespan assays in several mutant worms subjected to ND, HGD-1 and HGD-5. Upon ER stress, IRE-1 splices *xbp-1* mRNA (*xbp-1s*), resulting in the translation of the transcription factor XBP-1 which regulates a subset of UPR genes¹⁵. We included *xbp-1(lof)* animals as well as mutant animals with constitutive expression of *xbp-1s* in intestinal (*gly-19p::xbp-1s*) and neuronal (*rab-3p::xbp-1s*) cells¹⁶. A constitutive expression of *xbp-1s* in specific tissues translates into the upregulation of XBP-1 target genes regardless of ER stress. As an alternative to the IRE-1/XBP-1 axis, we also incorporated *jnk-1(lof)* animals. IRE-1 exhibits pro-apoptotic properties by phosphorylating c-Jun amino-terminal kinase (JNK-1) during chronic ER stress. As previously reported²⁰, the lifespan of *xbp-1(lof)* animals was shorter compared to WT on ND (Fig. 6g). The lifespan of *gly-19p::xbp-1s* animals was shorter while *rab-3p::xbp-1s* animals exhibited a significant increase in longevity when compared to WT on ND, similar to previous reports¹⁶⁻¹⁹. Finally, the lifespan of *jnk-1(lof)* animals on ND was slightly shorter compared to WT, validating previous findings^{24, 25}. The lifespan of *xbp-1(lof)* animals was exacerbated upon HGD-1 when compared to WT animals on HGD-1 while the difference in lifespans were minor between both strains on HGD-5 (Fig. 6h,i). This suggests that XBP-1 is necessary to extend the lifespan of aged animals on HGD while IRE-1 might hyperactivate the UPR through *xbp-1s* in certain tissues. In agreement with this hypothesis, the constitutive expression of *xbp-1s* in neuronal cells increased the lifespan of *rab-3p::xbp-1s* animals on HGD-1 and to a lesser extent on HGD-5 compared to ND. On the other hand, the constitutive expression of *xbp-1s* in intestinal cells had little effect on the lifespans of *gly-19p::xbp-1s* animals on HGD when compared to WT animals. Similarly, *jnk-1(lof)* animals exhibit similar lifespans on the different diets in comparison to WT animals. Conjointly with the transcriptomic analysis, these findings suggest that IRE-1 exacerbates the lifespan of HGD-1 animals, while ATF-6 and PEK-1 promote longevity of HGD-5 animals.”

As suggested by the reviewer, we quantified the expression of *far-3* gene in *ire-1(lof)* but there were no significant differences of *far-3* levels on ND, HGD-1, and HGD-5 when compared to WT. We only reported *far-3* levels in WT strains by using the *far-3p::GFP* reporter strain (Figure 3b).

6. The model argues that HGD at young age would result in chronic ER stress, which is toxic, while at late age, it increases UPR to promote longevity. This model would be strengthened if a model for constitutive UPRER is tested. Would an animal with constitutive UPRER be resistant to HGD-induced ER toxicity at young age? And would an animal with constitutive UPRER have no change from HGD at late age?

We thank the reviewer for the suggestions. We have addressed these questions in the comment # 5 above. Notably, the lifespan of *ire-1(lof)* animals subjected to HGD-1 was significantly longer than animals subjected to ND and HGD-5 (Figure 6f). These observations further support the model that IRE-1 is toxic to young animals subjected to HGD. However, the lifespans of animals with constitutive expression of *xbp-1s* in the intestinal or neuronal tissues were not in agreement with *ire-1(lof)* animals, suggesting that IRE-1 is detrimental to the lifespan of HGD-1 animals beyond the IRE-1/XBP-1 axis.

Minor comments:

-Last sentence in introduction (pg. 4) – did you mean to say “HGD was not dependent on *daf-16* (FOXO)?

We thank the reviewer for pointing this out. We rewrote the introduction and this misleading sentence has been removed.

-I am assuming that the *daf-2(e1370)* strain is used in this manuscript as that is the strain listed under the materials and methods section. This is one of the longest-lived mutants that exists, yet in Extended Data Fig. 2, the *daf-2(lof)* and WT on ND has the same lifespan. Can you comment on this? Also, *age-1* mutants exhibit extended lifespans in previous reports, yet a comparison of Figure 1a and Figure 2b suggests that you do not see a lifespan extension in *age-1* mutants. Please comment on this. Once again, a WT control would provide clarity here.

The reviewer is correct that we used *daf-2(e1370)*. We carried out the lifespan of *daf-2(lof)* several times as well as reordering *daf-2(e1370)* mutant from Caenorhabditis Genetics Center (CGC). In all cases, the lifespan of *daf-2(lof)* on normal diet was not as long as previously reported. Therefore, we felt that it was not worth reporting it and we decided to focus on *daf-16(lof)* lifespan.

Reviewer 3 comments

In the manuscript entitled “Glucose increases the lifespan of post-reproductive *C. elegans* independently of FOXO” authors provide evidence sustaining that a high glucose diet in aged nematodes can extend lifespan as opposed to young animals, where such regimen is a well-established inducer of lifespan shortening. Authors attempt to dissect the molecular bases of such findings by tackling the classical *daf-2/ daf-16* pathway, which was also shown to mediate lifespan regulation in previous publications from different groups. Additionally, authors aim to analyze the contribution of UPR sensors in high glucose diet mediated lifespan extension in aged organisms showing that ablating *atf-6* and *PEK* blocks high glucose diet effects on lifespan in aged nematodes, which was not observed in *IRE1* mutants. Authors perform RNAi screening followed by gene ontology and tissue enrichment analysis to understand how UPR mediates such effects and conclude that UPR plays a role in maintaining ER homeostasis in the intestine.

Overall, results are properly presented; however, for the standards of Nature Communication, authors were expected to evaluate deeply the molecular mechanisms behind lifespan extension in post-reproductive nematodes following HFD as this interesting result is of great interest to general scientific community studying metabolic alterations during aging. At least, a transcriptomic and/or proteomic analysis should be performed in different nematode tissues (the intestine and brain, for instance) to further understand what are the molecular basis of this observation. The contribution of UPR mediators to the phenomenon were not fully covered as conditional mutants of UPR mediators with expression in the intestine or brain tissue would clarify how ER stress response is transmitted from different tissues as described in the seminal 2013 paper of Rebeca Taylor and Dillin (doi:10.1016/j.cell.2013.05.042 (2013)).

Despite the interesting findings regarding lifespan extension in high glucose input in aged worms, I would not recommend this manuscript for publication in Nature Communications.

We thank the reviewer for an honest assessment of our original manuscript. We took the reviewer comments seriously and performed additional experiments to address some of the points. First, we carried out quantitative transcriptomic analysis in WT animals. (Fig. 2a,b). A single pool of synchronised worms was grown on ND until day 1 of adulthood (D1). From the same single pool, a subset of worms was fed on HGD at day 1 (HGD-1). The second subset of worms grown on ND was fed on HGD at day 5 (HGD-5) of adulthood (D5). Worms fed ND, HGD-1, and HGD-5 were harvested at D8 (n=6).

New Figure 2a,b:

The transcriptomic analysis guided us to further decipher the mechanisms driving lifespan extension of HGD-5 animals through the results presented in the new figures 2 to 6.

Next, we performed several additional lifespans of mutant animals related to the UPR. First, we extended the lifespans of *atf-6(lof)*, *ire-1(lof)*, and *pek-1(lof)* to HGD-1 (Figure 6d-f). Surprisingly, the lifespan of *ire-1(lof)* animals subjected to HGD-1 was significantly longer than animals subjected to ND and HGD-5 (Figure 6f). As suggested by the reviewer, we further characterise the role of IRE-1 pathway in modulating the longevity of animals subjected to HGD. The lifespan of *xbp-1(lof)* animals was shorter compared to WT on ND (Fig. 6g). The lifespan of *gly-19p::xbp-1s* (constitutive expression of spliced *xbp-1* in intestinal cells) animals was shorter while *rab-3p::xbp-1s* (constitutive expression of spliced *xbp-1* in neuronal cells) animals exhibited a significant increase in longevity when compared to WT on ND. Finally, the lifespan of *jnk-1(lof)* animals on ND was slightly shorter compared to WT. The lifespan of *xbp-1(lof)* animals was exacerbated upon HGD-1 when compared to WT animals on HGD-1 while the difference in lifespans were minor between both strains on HGD-5 (Fig. 6h,i). This suggests that XBP-1 is necessary to extend the lifespan of aged animals on HGD while IRE-1 might hyperactivate the UPR through *xbp-1s* in certain tissues. In agreement with this hypothesis, the constitutive expression of *xbp-*

1s in neuronal cells increased the lifespan of *rab-3p::xbp-1s* animals on HGD-1 and to a less extent on HGD-5 compared to ND. On the other hand, the constitutive expression of *xbp-1s* in intestinal cells had little effect on the lifespans of *gly-19p::xbp-1s* animals on HGD when compared to WT animals. Similarly, *jnk-1(lof)* animals exhibit similar lifespans on the different diets in comparison to WT animals. Conjointly with the transcriptomic analysis, these findings suggest that IRE-1 exacerbates the lifespan of HGD-1 animals, while ATF-6 and PEK-1 promote longevity of HGD-5 animals.

New Figure 6d-i:

Updated manuscript (lines 268-298)

“Hereafter, we asked if the UPR renders a role in extending the lifespan of HGD-5 animals. The lifespans of *atf-6(lof)* and *pek-1(lof)* animals on HGD from D1 or D5 were similar to ND (Fig. 6d,e, Supplementary Table 1). These findings indicate that both ATF-6 and PEK-1 participate in extending the longevity of D5 worms on HGD. In contrast, HGD extended the lifespan of D5 *ire-1(lof)* animals when compared to ND (Fig. 6f, Supplementary Table 1). Surprisingly, the lifespan of *ire-1(lof)* animals on HGD-1 was longer than *ire-1(lof)* animals on ND and HGD-5 as well as WT animals on ND (Supplementary Fig. 4d and Table 1).

To further characterise the role of IRE-1 pathway in modulating the longevity of animals subjected to HGD, we conducted lifespan assays in several mutant worms subjected to ND, HGD-1 and HGD-5. Upon ER stress, IRE-1 splices *xbp-1* mRNA (*xbp-1s*), resulting in the translation of the transcription factor XBP-1 which regulates a subset of UPR genes¹⁵. We included *xbp-1(lof)* animals as well as mutant animals with constitutive expression of *xbp-1s* in intestinal (*gly-19p::xbp-1s*) and neuronal (*rab-3p::xbp-1s*) cells¹⁶. A constitutive expression of *xbp-1s* in specific tissues translates into the upregulation of XBP-1 target genes regardless of ER stress. As an alternative to the IRE-1/XBP-1 axis, we also incorporated *jnk-1(lof)* animals. IRE-1 exhibits pro-apoptotic properties by phosphorylating c-Jun amino-terminal kinase (JNK-1) during chronic ER stress. As previously reported²⁰, the lifespan of *xbp-1(lof)* animals was shorter compared to WT on ND (Fig. 6g). The lifespan of *gly-19p::xbp-1s* animals was shorter while *rab-3p::xbp-1s* animals exhibited a significant increase in longevity when compared to WT on ND, similar to previous reports^{16, 19}. Finally, the lifespan of *jnk-1(lof)* animals on ND was slightly shorter compared to WT, validating previous findings^{64, 65}. The lifespan of *xbp-1(lof)* animals was exacerbated upon HGD-1 when compared to WT animals on HGD-1 while the difference in lifespans were minor between both strains on HGD-5 (Fig. 6h,i). This suggests that XBP-1 is necessary to extend the lifespan of aged animals on HGD while IRE-1 might hyperactivate the UPR through *xbp-1s* in certain tissues. In agreement with this hypothesis, the constitutive expression of *xbp-1s* in neuronal cells increased the lifespan of *rab-3p::xbp-1s* animals on HGD-1 and to a less extent on HGD-5 compared to ND. On the other hand, the constitutive expression of *xbp-1s* in intestinal cells had little effect on the lifespans of *gly-19p::xbp-1s* animals on HGD when compared to WT animals.

Similarly, *jnk-1(lof)* animals exhibit similar lifespans on the different diets in comparison to WT animals. Conjointly with the transcriptomic analysis, these findings suggest that IRE-1 exacerbates the lifespan of HGD-1 animals, while ATF-6 and PEK-1 promote longevity of HGD-5 animals.”

Minor comments:

1. Did authors measure levels of *daf-2* and *age-1* following HGD? Do they go to the nucleus? Authors performed this assay for *daf-16* but not for *daf-2* and *age-1*. What happens with *daf-16* mutants if submitted to the same unconventional temperature shift implemented to *daf-2* evaluation?

We thank the reviewer for the suggestions. The highly conserved insulin/IGF-1 signaling (IIS) pathway consists of the insulin-like growth factor 1 receptor (DAF-2) which is a transmembrane protein at the plasma membrane. Activated DAF-2 induces a cascade of kinases through AGE-1 resulting in the phosphorylation and inactivation of FOXO transcription factor DAF-16 by inhibiting its translocation to the nucleus. On the other hand, quiescent DAF-2 promotes the localization of DAF-16 to the nucleus which in turn upregulate genes related to hormones, detoxification, anti-inflammation, metabolism, and lipolysis. Unfortunately, there is no commercial antibodies to monitor the phosphorylated states of each protein of the DAF-2/DAF-16 pathway. By convention in *C. elegans*, the research community has relied on the localisation of DAF-16 tagged to GFP (DAF-16::GFP) to monitor the activity status of the DAF-2/DAF-16 pathway.

We haven't tried to submit *daf-16(lof)* to the same unconventional temperature shift implemented to *daf-2(lof)* but we have done so for WT. We anticipate that no difference will be observed at unconventional temperature and normal temperature for *daf-16(lof)* as the mutant does not form dauer.

2. Authors should confirm the finding that ATF-6 and PEK activation compensate for weak activation of IRE1 by treating worms with tunicamycin. Would IRE1 show an increased disruption compared to *atf6* and *pek* activation? if it does not, maybe the findings in HGD regarding UPR activation are specific to IRE1 pathway in the context of glucose metabolism.

We thank the reviewer for the suggested experiment. Low concentration of tunicamycin will indeed mildly activate the UPR through the three ER stress sensors ATF-6, PEK-1, and IRE-1. Tunicamycin inhibits overall glycosylation of ER resident/secreted proteins, resulting in an accumulation of misfolded proteins at the ER. Therefore, long term exposure to tunicamycin will affect many pathways and processes unrelated to ER stress. As an alternative, we opted to use the constitutively activated IRE-1 pathway from Rebeca Taylor and Dillin labs (PMID 23791175). We carried out the lifespan assays of *gly-19::xbp-1s* (IRE-1 branch constitutively activated in the intestine) and *rab-3::xbp-1s* (IRE-1 branch constitutively activated in the neurons). We reported our findings in the section Reviewer #3 general comments.

3. It is interesting to note that authors sustain that IRE1 does not have a role in extending lifespan following HGD, however, Taylor paper shows that XBP1s is necessary to increase lifespan by cell non autonomous responses in the intestine in a normal context. How do authors interpret this data?

We thank the reviewer for raising this issue. Perhaps, our rationale was not eloquently described in the original submission and that we missed the supporting data to conclude that IRE-1 was not implicated in extending the lifespan of worms fed HGD. We addressed these questions extensively in our reply to review #3 above. Notably, the lifespan of *ire-1(lof)* animals subjected to HGD-1 was significantly longer than animals subjected to ND and HGD-5 (Figure 6f). These observations further support the model that IRE-1 is toxic to young animals subjected to HGD. However, the lifespans of animals with constitutive expression of *xbp-1s* in the intestinal or neuronal tissues were not in agreement with *ire-1(lof)* animals, suggesting that IRE-1 is detrimental to the lifespan of HGD-1 animals beyond the IRE-1/XBP-1 axis.

Reviewers' Comments:

Reviewer #1:

Remarks to the Author:

Review of revised manuscript

The authors have done a very nice job in trying to address all of the concerns. However, this reviewer still does not support publishing this manuscript. My major concerns are listed below. Primarily, the fact that all of the data relies like the FUDR is of major concern. Rather than the effect of glucose, this manuscript is the effect of the FUDR.

1-"In the absence of FUDR, the lifespan of WT animals on ND was comparable to those grown in the presence of FUDR (Fig. 4b, Supplementary Table 1), in agreement with previous reports⁴² , ⁴³. Similarly, animals on HGD-1 exhibited a shorter lifespan compared to ND. Conversely, the lifespan of HGD-5 animals was drastically shortened in the absence of FUDR and comparable to the lifespan of animals on HGD-1. "

This suggests all of the data is simply a response to the drug FUDR. In fact, there is a large literature about the effects of this drug on lifespan and gene expression. See below: BMC Res Notes 2021 May 28;14(1):207. doi: 10.1186/s13104-021-05624-6. Effects of FUDR on gene expression in the *C. elegans* bacterial diet OP50 McIntyre et al.

2-'To further assess the role of fertility in modulating lifespan, we used long-lived germline defective *glp-1*(loss-of-function; *lof*) animals⁴⁵. Surprisingly, the lifespan of *glp-1*(*lof*) animals fed on HGD by day 5 was significantly shorter compared to HGD-1 in the absence of FUDR (Fig. 4c, Supplementary Table 1). Together, these findings suggest that germline proliferation plays a role in promoting the longevity of HGD-5 animals.'

The authors do not consider the other possibility that there is simply a change in gene expression due to FUDR. All of the data is dependent on FUDR not glucose!

Reviewer #2:

Remarks to the Author:

The authors have sufficiently addressed my concerns with their revision.

Reviewer #3:

Remarks to the Author:

The manuscript has been properly revised and now deserves publication in Nat Com

Reviewer #4:

Remarks to the Author:

The authors use *C. elegans* to interrogate the effects of glucose on longevity pre- and post-reproduction and the role that ER stress pathways play in regulating these effects on longevity. The manuscript is improved, but still possesses some major flaws. If the comments below can be addressed adequately I recommend acceptance into Nature Communications.

Major comments:

Comment 1: The title of the manuscript does not adequately describe the results. The word "aging", "lifespan", or "longevity" should be present. The title might also reflect the important finding that germ cell proliferation regulates the glucose-induced longevity effect in aged *C.*

elegans. A suggestion is "Glucose extends lifespan when added to aged FUDR-sterilized *C. elegans* dependent upon the ER stress sensor IRE-1".

Comment 2: As another reviewer mentioned, it is problematic that you show the *glp-1(lof)* worms have a shorter lifespan than the controls when they are well-known to be a long-lived strain. On line 730 the strain name is wrong. Instead of *glp-1(cf1903)* it should be *glp-1(e2144)*. No meaningful conclusions can be drawn from those sets of experiments. The finding that HGD-5 does not extend lifespan in *glp-1* worms is also not that surprising given that this strain already likely has longevity pathways activated. Due to these factors, it would greatly strengthen the results to determine the effects of HGD-1 and HGD-5 on lifespan in normal-lived temperature-sensitive sterile worm strains such as female *fem-1(hc17)* or hermaphrodite *glp-4(bn2)* when grown at 25°C. Experiments with the *fem-1(hc17)* strain could suggest if egg precursor cells are required for the HGD-5-induced longevity effect.

Comment 3: The lowered number of *E. coli* present in the digestive tract of HGD-5 worms might suggest that protein restriction could be contributing to the longevity. Experiments addressing this possibility are warranted. It is somewhat counterintuitive that the pharyngeal pumping rate doesn't change but the number of *E. coli* present in the digestive tract changes with HGD-5. Could fewer *E. coli* be consumed with each pharyngeal pump with HGD-5? Are there increased levels of proteases in the gut to degrade GFP during HGD-5? Or does the rate of peristalsis and defecation increase to cause the decreased number of *E. coli* present in the digestive tract? If there is less protein being taken up from the intestine, does the HGD-5-induced longevity depend upon *pha-4* that is required for dietary restriction-induced longevity? Does amino acid/protein supplementation prevent the lifespan extension? This is important as a good protocol to study the effects of protein restriction on aging in *C. elegans* has yet to be developed.

Comment 4: line 55 in abstract and lines 101 and 181: fertility -> infertility [FUDR-induced infertility is required for HGD-5-induced longevity]

Comment 5: Please cite the recent paper and compare and contrast your results to theirs. Protective Effects of Transient Glucose Exposure in Adult *C. elegans*. Murillo K, Samigullin A, Humpert PM, Fleming T, Özer K, Schlotterer A, Hammes HP, Morcos M. Antioxidants (Basel). 2022 Jan 14;11(1):160. doi: 10.3390/antiox11010160. PMID: 35052664.

Also, it is important to mention that glucose can extend lifespan when only present during the larval stages, so the reproductive stage appears to be where glucose is most detrimental. Fragile lifespan expansion by dietary mitohormesis in *C. elegans*. Tauffenberger A, Vaccaro A, Parker JA. Aging (Albany NY). 2016 Jan;8(1):50-61. doi: 10.18632/aging.100863. PMID: 26764305

Comment 6: It is important to discuss that FUDR kills or at least halts cell division of all mitotic germ cells starting at the L4 stage when it is added, while the *glp-1* mutation only interferes with the cell division of a subset of the mitotic germ cells but beginning from the egg and sperm, thus explaining the different effects of HGD-5 in these two different conditions.

Comment 7: It is unclear in Supp Fig3B which p-value goes with each comparison. Please make it clearer which p-value goes with ND vs. HGD-1 and which one goes with ND vs. HGD-5. I assume from the text that the p-value of 0.5298 is the comparison between HGD-1 and HGD-5 and the $p < 0.001$ go with the other two.

Comment 8: The explanation for the brood size experiment in Supp Fig 3C is not clear. Why should there be any brood at all if FUDR is present from L4 to adult day 7 and eggs are normally laid from adult days 1 through 5? Is egg-laying delayed when FUDR is present and so all of the eggs are laid after adult day 7 to worms on ND? Are worms on HGD-1 unable to delay their egg-laying until day 8 and after and therefore have a much smaller brood? If so, please explain this more clearly. In the legend also indicate that filled circles represent FUDR is present and open circles indicate FUDR is absent.

Comment 9: On Line 219 it is written "findings suggest that embryogenesis counteracts life extension induced by HGD in aged animals." No brood size experiments are shown for HGD-5 animals, just on ND and HGD-1 animals. Therefore, this statement is inaccurate and could be

changed to "findings suggest that HGD in young animals decreases both the brood size and lifespan".

Comment 10: Line 330 It has already been shown that calorie restriction when initiated later in life can extend lifespan of several organisms including mice (although not to the extent it can when it is initiated earlier).

Minor comments: wording

Line 197: by -> starting on

Line 199: promoting -> regulating

Line 209: promote -> promotes

Line 214: days 7 -> day 7

Line 241: from extending -> from the nucleus and is not involved in extending

Line 257: are -> are likely

Line 268: renders -> plays

Line 276: of IRE-1 -> of the IRE-1

Line 290: suggest -> suggests

Line 322: sapsins -> saposins [perhaps I don't know what a sapsin is]

Line 329: combined -> when combined

Line 337: calorie restriction increases -> some dietary restriction protocols increase

Line 341 and 342: I would remove the words "to extend the lifespan of aged animals" [unfounded speculation]

Line 346: animal -> animals

Line 361: fitness -> relative function

Line 366: ATF-6 and PEK-1 while attenuating IRE-1 pathways -> ATF-6 and PEK-1 pathways while attenuating the IRE-1 pathway

Line 368: in preventing -> to prevent

REPLY TO THE REVIEWERS

We are grateful for the positive overall assessment of our revised work by all four reviewers. Critical comments on specific aspects of our work are listed verbatim below followed by our responses.

Reviewer 1 comments

The authors have done a very nice job in trying to address all of the concerns. However, this reviewer still does not support publishing this manuscript. My major concerns are listed below. Primarily, the fact that all of the data relies like the FUDR is of major concern. Rather than the effect of glucose, this manuscript is the effect of the FUDR.

We thank the reviewer to recognise the efforts we have made to address the reviewers' concerns.

1. "In the absence of FUDR, the lifespan of WT animals on ND was comparable to those grown in the presence of FUDR (Fig. 4b, Supplementary Table 1), in agreement with previous reports^{42, 43}. Similarly, animals on HGD-1 exhibited a shorter lifespan compared to ND. Conversely, the lifespan of HGD-5 animals was drastically shortened in the absence of FUDR and comparable to the lifespan of animals on HGD-1. "

This suggests all of the data is simply a response to the drug FUDR. In fact, there is a large literature about the effects of this drug on lifespan and gene expression. See below:
BMC Res Notes 2021 May 28;14(1):207. doi: 10.1186/s13104-021-05624-6. Effects of FUDR on gene expression in the *C. elegans* bacterial diet OP50 McIntyre et al.

2. 'To further assess the role of fertility in modulating lifespan, we used long-lived germline defective *glp-1*(loss-of-function; *lof*) animals⁴⁵. Surprisingly, the lifespan of *glp-1*(*lof*) animals fed on HGD by day 5 was significantly shorter compared to HGD-1 in the absence of FUDR (Fig. 4c, Supplementary Table 1). Together, these findings suggest that germline proliferation plays a role in promoting the longevity of HGD-5 animals.'

The authors do not consider the other possibility that there is simply a change in gene expression due to FUDR. All of the data is dependent on FUDR not glucose!

We agree with the reviewer that the role of FUDR in modulating the lifespan is ambiguous as highlighted by the literature cited by the reviewer above. We addressed this point through the manuscript and clearly articulated our points in the rebuttal. It was important to us to provide as much details as possible as we are strong advocate of being fully transparent and honest in revealing our findings.

In the revised manuscript (NCOMMS-18-20231A), our new data reported in Figure 4b-e demonstrate that FUDR has a synergic effect with glucose to modulate the lifespan of hermaphrodite animals. On the other hand, FUDR has no effect on HGD-modulated longevity of male animals. To minimize the effects of FUDR on the lifespan of hermaphrodite animals, we subjected the animals to FUDR from L4 to D8 only. The lifespan of HGD-5 was significantly extended with the minimal FUDR exposure from L4 to D8 (Figure 4e). Therefore, we strongly argue that glucose significantly increases the lifespan of high glucose diet at day 5 of adulthood (HGD-5) compared to normal diet (ND) which might be partially dependent of FUDR through the reproductive period.

FUDR is commonly used to conduct the lifespan of *C. elegans*. Some research groups removed FUDR from the agar plate media (NMG plates) once the animals passed the reproduction period. However, many groups prefer to keep FUDR through the entire adulthood for consistency. Here are some examples of **reported lifespans assay with FUDR** published in recent years:

1. D. J. Cattie, ... D. H. Kim (2016). Mutations in Nonessential eIF3k and eIF3l Genes Confer Lifespan Extension and Enhanced Resistance to ER Stress in *Caenorhabditis elegans*. PLOS Genetics, September 30, 2016 <https://doi.org/10.1371/journal.pgen.1006326>

"Lifespan assays were carried out as previously described [52]. Briefly, 30 L4 worms were transferred to **NGM plates containing 50 µg/ml-1 5-fluoro-2'-deoxyuridine (FUDR)** in triplicate and the assay was carried out at 25°C."

2. A. L. Chen, ... B.F. Cravatt (2019). Pharmacological convergence reveals a lipid pathway that regulates *C. elegans* lifespan. Nature Chemical Biology 15, pages453–462 (2019) <https://doi.org/10.1038/s41589-019-0243-4>

"In brief, approximately ten age-synchronized *C. elegans* were cultured at 20 °C in 96-well plates containing S-complete media and irradiated Op50 bacteria (6 mg/mL). **Animals, except the *glp-1***

strain, were given FUDR (0.6 mM, Sigma) at the L4 developmental stage to prevent the development of progeny.”

3. Wan Q, Meng X, Fu X, Chen B, Yang J, Yang H, Zhou Q. Intermediate metabolites of the pyrimidine metabolism pathway extend the lifespan of *C. elegans* through regulating reproductive signals. *Aging* (Albany NY). 2019; 11:3993-4010. <https://doi.org/10.18632/aging.102033>
“*In brief, 100 late L4 larvae or young adults were transferred to fresh plates containing 10 μM 5-fluoro-2'-deoxyuridine (FUdR, Sigma) and the respective compounds and scored every day.*”
4. Essmann, C.L., Martinez-Martinez, D., Pryor, R. et al. Mechanical properties measured by atomic force microscopy define health biomarkers in ageing *C. elegans*. *Nat Commun* 11, 1043 (2020). <https://doi.org/10.1038/s41467-020-14785-0>
“*At day 1 of adulthood, worms were transferred to each of the treatment/condition plates containing FUdR (50 μM) and transferred every 4 days to new fresh plates until day 12.*”
5. Marina Ezcurra, Alexandre Benedetto, Thanet Sornda, Ann F. Gilliat, Catherine Au, Qifeng Zhang, Sophie van Schelt, Alexandra L. Petrache, Hongyuan Wang, Yila de la Guardia, Shoshana Bar-Nun, Eleanor Tyler, Michael J. Wakelam, David Gems, *C. elegans* Eats Its Own Intestine to Make Yolk Leading to Multiple Senescent Pathologies, *Current Biology*, Volume 28, Issue 16, 2018, <https://doi.org/10.1016/j.cub.2018.06.035>.
“*Worms were either transferred daily during the reproductive period, or transferred at L4 stage to plates supplemented with 15 μM FUDR to block progeny production*”
6. Liu W, Lin H, Mao Z, Zhang L, Bao K, Jiang B, Xia C, Li W, Hu Z, Li J, . Verapamil extends lifespan in *Caenorhabditis elegans* by inhibiting calcineurin activity and promoting autophagy. *Aging* (Albany NY). 2020; 12:5300-5317. <https://doi.org/10.18632/aging.102951>
“*In addition, 50 μg/mL of 5-Fluorodeoxyuridine (FudR) was added to the agar plates from day 0 to day 10 to avoid progeny hatching.*”
7. Franco-Juárez B, Mejía-Martínez F, Moreno-Arriola E, Hernández-Vázquez A, Gómez-Manzo S, Marcial-Quino J, Arreguín-Espinosa R, Velázquez-Arellano A, Ortega-Cuellar D, . A high glucose diet induces autophagy in a HLH-30/TFEB-dependent manner and impairs the normal lifespan of *C. elegans*. *Aging* (Albany NY). 2018; 10:2657-2667. <https://doi.org/10.18632/aging.101577>
“*... the worms were moved to NGM control plates or to glucose-supplemented plates (100 mM) previously seeded with E. coli OP50-1 and supplemented with 49 μM of 5-fluoro-2'-deoxyuridine (FUDR, Sigma-Aldrich).*”
8. Cornwell A.B., Samuelson A.V. (2020) Analysis of Lifespan in *C. elegans*: Low- and High-Throughput Approaches. In: Curran S. (eds) *Aging. Methods in Molecular Biology*, vol 2144. Humana, New York, NY. https://doi.org/10.1007/978-1-0716-0592-9_2
“*... when animals reach the L4 stage dilute filter-sterile 1000× FUdR stock to 160× with ultrapure water and add 50 μL of 160× FUdR to the center of each 60 mm plate*”
9. Puchalt, J.C., Sánchez-Salmerón, A.J., Ivorra, E. et al. Small flexible automated system for monitoring *Caenorhabditis elegans* lifespan based on active vision and image processing techniques. *Sci Rep* 11, 12289 (2021). <https://doi.org/10.1038/s41598-021-91898-6>
“*FUdR (0.2 mM) was used to prevent reproduction, and fungizone (1μg/mL) was added to prevent fungal contaminations*”
10. Yonghak Seo, Samuel Kingsley, Griffin Walker, Michelle A. Mondoux, Heidi A. Tissenbaum. Metabolic shift from glycogen to trehalose promotes lifespan and healthspan in *Caenorhabditis elegans*. *Proceedings of the National Academy of Sciences* Mar 2018, 115 (12) E2791-E2800; DOI: 10.1073/pnas.1714178115
“*Then, ~100 L4s were transferred to NGM or RNAi plates containing 200 μM FUdR (Pharma Waldhof GmbH), with/without added sugar and kept at 20 °C (~33 animals per plate).*”

Therefore, we strongly argue that the difference in lifespan we observed between ND, HGD-1, and HGD-5 are biologically relevant and that they depend on the unfolded protein response (UPR). However, we cannot exclude that FUdR might play a partial synergic role with glucose in modulating the lifespan of animals on HGD.

FUdR is commonly used to obtain large scale synchronized *C. elegans*. To perform the RNA-seq assay, animals were harvested at day 8 of adulthood in the presence of FUdR. As illustrated above, the use of FUdR is not unusual especially during the reproductive phase of the animals (up to ~ day 7 of adulthood). The use of FUdR is even more critical when large populations of synchronised animals are required such as for RNA-seq, proteomics, immunoblot, quantitative PCR, and assays to quantify metabolites. Here are some examples of **reported RNA-seq or proteomic assays of aged animals with FUdR** published in recent years:

1. Koyuncu, S., Loureiro, R., Lee, H.J. et al. Rewiring of the ubiquitinated proteome determines ageing in *C. elegans*. *Nature* 596, 285–290 (2021). <https://doi.org/10.1038/s41586-021-03781-z>
“*To obtain large populations of synchronized hermaphrodite... Then, worms were transferred onto plates with OP50 E. coli (covered with 100 μg ml⁻¹ 5-fluoro-*

2'deoxyuridine (FUdR) to prevent the development of progeny. Every five days, adult worms were transferred onto fresh plates."

2. Gusarov, I., Shamovsky, I., Pani, B. et al. Dietary thiols accelerate aging of *C. elegans*. *Nat Commun* 12, 4336 (2021). <https://doi.org/10.1038/s41467-021-24634-3>
"To study transcriptional response to NAC wt worms were allowed to develop and grow on LB, DB or DB + 15 mM NAC agar plates at 20 °C **until day 8 of adulthood. Heat-inactivated bacteria were used to prepare DB agar plates. To avoid extensive internal hatching 40 µM FUdR was added after the worms reached L4 stage.**"
3. Narayan V, Ly T, Pourkarimi E, Murillo AB, Gartner A, Lamond AI, Kenyon C. Deep Proteome Analysis Identifies Age-Related Processes in *C. elegans*. *Cell Syst*. 2016 Aug;3(2):144-159. doi: 10.1016/j.cels.2016.06.011.
"... when the worms were **mid-late L4 animals**, they were washed off the plates with M9 buffer and transferred to fresh **NGM-N plates containing 50 µM FUdR.**"
4. Huang, Q., Li, R., Yi, T. et al. Phosphorothioate-DNA bacterial diet reduces the ROS levels in *C. elegans* while improving locomotion and longevity. *Commun Biol* 4, 1335 (2021). <https://doi.org/10.1038/s42003-021-02863-y>
"Briefly, worms were placed on **mNGM plates supplemented with 0.1 mM of FUdR** and covered with lawns of OP50... Age-synchronized nematodes were prepared as described above."
5. Heissenberger C, Rollins JA, Krammer TL, Nagelreiter F, Stocker I, Wacheul L, Shpylovyi A, Tav K, Snow S, Grillari J, Rogers AN, Lafontaine DLJ, Schosserer M. The ribosomal RNA m5C methyltransferase NSUN-1 modulates healthspan and oogenesis in *Caenorhabditis elegans*. *Elife*. 2020 Dec 8;9:e56205. doi: 10.7554/eLife.56205
"... **young adult worms per condition were placed on fresh NGM plates containing 5 mL NGM, 100 µL bacterial suspension and 50 µg FUdR.**"

Taken together, our approach of using FUdR to prevent the mixture of progenies with adult through the reproductive period of the animal is common to conduct RNA-seq analysis. We strongly argue that our transcriptomic data is supported by the different assays conducted on animals that were exposed to FUdR until harvest at day 6 to day 12 of adulthood. Moreover, several of our findings are supported by the literature.

Reviewer 2 comments

The authors have sufficiently addressed my concerns with their revision.

We are delighted that we addressed Reviewer #2 comments appropriately.

Reviewer 3 comments

The manuscript has been properly revised and now deserves publication in Nat Com

We are delighted that we addressed Reviewer #3 comments appropriately.

Reviewer 4 comments

The authors use *C. elegans* to interrogate the effects of glucose on longevity pre- and post-reproduction and the role that ER stress pathways play in regulating these effects on longevity. The manuscript is improved, but still possesses some major flaws. If the comments below can be addressed adequately I recommend acceptance into Nature Communications.

1. The title of the manuscript does not adequately describe the results. The word "aging", "lifespan", or "longevity" should be present. The title might also reflect the important finding that germ cell proliferation regulates the glucose-induced longevity effect in aged *C. elegans*. A suggestion is "Glucose extends lifespan when added to aged FUdR-sterilized *C. elegans* dependent upon the ER stress sensor IRE-1".

We thank the reviewer for the suggestion. We modified the title to "The unfolded protein response modulates lifespan in chemically-sterilised *C. elegans* upon glucose diet"

2. As another reviewer mentioned, it is problematic that you show the *glp-1(lof)* worms have a shorter lifespan than the controls when they are well-known to be a long-lived strain. On line 730 the strain name is wrong. Instead of *glp-1(cf1903)* it should be *glp-1(e2144)*. No meaningful conclusions can be

drawn from those sets of experiments. The finding that HGD-5 does not extend lifespan in *glp-1* worms is also not that surprising given that this strain already likely has longevity pathways activated. Due to these factors, it would greatly strengthen the results to determine the effects of HGD-1 and HGD-5 on lifespan in normal-lived temperature-sensitive sterile worm strains such as female *fem-1(hc17)* or hermaphrodite *glp-4(bn2)* when grown at 25°C. Experiments with the *fem-1(hc17)* strain could suggest if egg precursor cells are required for the HGD-5-induced longevity effect.

We thank the reviewer for pointing this out. We have carried out the lifespan of *fem-1(hc17)* fed ND, HGD-1, and HGD-5. As expected, *fem-1(lof)* animals lived longer than WT on normal diet at 25°C (Supplementary Fig. S3a and Table 1). The lifespan of *fem-1(lof)* animals fed on HGD, at the equivalent of 20°C-day-5 at 25°C, was significantly shorter compared to ND in the absence of FUdR (Fig. 4c, Supplementary Fig. S3a and Table 1). Similarly, aged germline defective *glp-1(lof)* animals exhibited a shorter lifespan on HGD (Supplementary Fig. S3b and Table 1).

New Figure 4c:

New Supplementary Figure 3a,b:

Updated manuscript (lines 203-212)

“To further assess the role of fertility in modulating lifespan, we used long-lived germline defective *fem-1* (loss-of-function; *lof*) animals⁴⁶. In contrast to FUdR, which prevents cell division indiscriminately when subjecting larva 4 stage animals, *fem-1* mutation results in a loss of the stem cell population that is precursor to gamete cells⁴⁷. As expected, *fem-1(lof)* animals lived longer than WT on normal diet at 25°C (Supplementary Fig. S3a and Table 1). Surprisingly, the lifespan of *fem-1(lof)* animals fed on HGD, at the equivalent of day 5 at 25°C, was significantly shorter compared to ND in the absence of FUdR (Fig. 4c, Supplementary Table 1). Similarly, aged germline defective *glp-1(lof)* animals exhibited a shorter lifespan on HGD (Supplementary Fig. S3b and Table 1). Together, these findings suggest that germline proliferation plays a role in regulating the longevity of HGD-5 animals.”

3. The lowered number of *E. coli* present in the digestive tract of HGD-5 worms might suggest that protein restriction could be contributing to the longevity. Experiments addressing this possibility are warranted. It is somewhat counterintuitive that the pharyngeal pumping rate doesn't change but the number of *E. coli* present in the digestive tract changes with HGD-5. Could fewer *E. coli* be consumed with each pharyngeal pump with HGD-5? Are there increased levels of proteases in the gut to degrade GFP during HGD-5? Or does the rate of peristalsis and defecation increase to cause the decreased number of *E. coli* present in the digestive tract? If there is less protein being taken up from the intestine, does the HGD-5-induced longevity depend upon *pha-4* that is required for dietary restriction-induced longevity? Does amino acid/protein supplementation prevent the lifespan extension? This is important as a good protocol to study the effects of protein restriction on aging in *C. elegans* has yet to be developed.

We agree with the reviewer that our data from bacteria intake and bacteria present in the digestive track are inconsistent. To address this issue, we have carefully considered both assays by including HGD-1 as it was missing in our reported data.

We ran several trial of measuring OP50-GFP levels in the digestive tracks of WT fed ND, HGD-1, or HGD-5 at day 8 and day 11 of adulthood. For this assay, animals are harvested and carefully washed

to remove any OP50-GFP that are attached to the animal cuticle. This process takes about 15 minutes. Unfortunately, we noticed a wide variability in the levels of OP50-GFP in the digestive tracks of the animals. This might be since the animals are old. Moreover, the levels of autofluorescence were significant, making quantification challenging. In our previous submission, OP5-GFP levels in the digestive tracks were measure only after 24h exposure to HGD.

Therefore, we focused on measuring the levels of bacteria at OD₆₀₀ in liquid media. WT animals were grown on solid NGM and fed ND, HGD-1, and HGD-5. At day 5 of adulthood, animals were transferred to liquid media containing OP50. The microplates were incubated 3 days at 20°C and the OP50 density in liquid culture was measured (Fig. 1d). Bacteria intake was significantly higher for HGD-1 but not for HGD-5 compared to ND. This is consistent with the pumping rates that are similar between ND and HGD-5. In our previous bacteria intake experiment, we reported that bacteria intake was lower in HGD-5 compared to ND. However, OP50 density was measured only 24h after exposing the animals to HGD-5. We argue that measuring OP50 density after 3 days exposure to HGD-5 is a better reflection of the animal food intake as the animals might need a period of adaptation to the stress of HGD. Additionally, it was reported that OP50 density in the presence of worms is greater and consistent compared to OP50 alone after 3 or 4 days (PMID 25903497).

New Figure 1d:

Updated manuscript (lines 131-134)

"To assess whether HGD influences food intake in D5 worms, we measured the change of OP50 density in liquid culture after 3 days incubation (Figure 1d). In WT worms, bacteria levels were significantly higher in HGD-1 while being similar in HGD-5 when compared to ND."

4. line 55 in abstract and lines 101 and 181: fertility -> infertility [FUdR-induced infertility is required for HGD-5-induced longevity]

We agree that using "fertility" was misleading. We modified it accordingly as well as in the title of Figure 4.

Updated manuscript

"We observed a metabolic shift only in HGD-1, while glucose and infertility synergistically prolonged the lifespan of HGD-5, independently of DAF-16." (lines 64-65)

"A metabolic shift was observed in HGD-1 while glucose and infertility synergistically prolonged the lifespan of HGD-5, independently of DAF-16." (lines 110-111)

"Glucose and infertility synergistically prolong lifespans of aged animals" (line 188 and 442)

5. Please cite the recent paper and compare and contrast your results to theirs. Protective Effects of Transient Glucose Exposure in Adult *C. elegans*. Murillo K, Samigullin A, Humpert PM, Fleming T, Özer K, Schlotterer A, Hammes HP, Morcos M. *Antioxidants* (Basel). 2022 Jan 14;11(1):160. doi: 10.3390/antiox11010160. PMID: 35052664.

Also, it is important to mention that glucose can extend lifespan when only present during the larval stages, so the reproductive stage appears to be where glucose is most detrimental. Fragile lifespan expansion by dietary mitohormesis in *C. elegans*. Tauffenberger A, Vaccaro A, Parker JA. *Aging* (Albany NY). 2016 Jan;8(1):50-61. doi: 10.18632/aging.100863. PMID: 26764305

We agree with the reviewer that incorporating these two publications to our discussion is important. We have modified the first paragraph of the discussion section accordingly.

Updated manuscript (lines 324-331)

"Our study points to a specific life stage from which it could be beneficial to promote artificial proteostasis and subsequently attenuate the consequences of ageing. Interestingly, animals subjected to HGD at the pre-reproductive stage or short HGD exposure in young adults have been reported to extend lifespan^{72, 73}. The timing and length for intervention might be critical, as it could exacerbate ageing if done unduly early and persistently. It remains to be interrogated whether the treatment that activates the UPR in post-reproductive mammals correlates with an extension of lifespan through the clearance of intracellular damages."

6. It is important to discuss that FUdR kills or at least halts cell division of all mitotic germ cells starting at the L4 stage when it is added, while the *glp-1* mutation only interferes with the cell division of a subset of the mitotic germ cells but beginning from the egg and sperm, thus explaining the different effects of HGD-5 in these two different conditions.

We thank the reviewer for giving us an opportunity to clarify the differences between FUdR and *glp-1* mutation on the maturation of germline stem cells. We have modified the manuscript accordingly.

Updated manuscript

"FUdR inhibits germline stem cell proliferation and the production of intact eggs in adults⁴⁰." (lines 196-197)

*"In contrast to FUdR which prevents cell division indiscriminately when subjecting stage 4 larval animals, *fem-1* mutation results in a loss of stem cell population which is precursor to gamete cells⁴⁷." (lines 205-207)*

7. It is unclear in Supp Fig3B which p-value goes with each comparison. Please make it clearer which p-value goes with ND vs. HGD-1 and which one goes with ND vs. HGD-5. I assume from the text that the p-value of 0.5298 is the comparison between HGD-1 and HGD-5 and the $p < 0.001$ go with the other two.

We modified the *P* value labelling for the lifespan graph in Supplementary Fig. 3d (previously 3b) as well as the other lifespan graphs through the figures for consistency.

New Supplementary Figure S3c,d:

8. The explanation for the brood size experiment in Supp Fig 3C is not clear. Why should there be any brood at all if FUdR is present from L4 to adult day 7 and eggs are normally laid from adult days 1 through 5? Is egg-laying delayed when FUdR is present and so all of the eggs are laid after adult day 7 to worms on ND? Are worms on HGD-1 unable to delay their egg-laying until day 8 and after and therefore have a much smaller brood? If so, please explain this more clearly. In the legend also indicate that filled circles represent FUdR is present and open circles indicate FUdR is absent.

We thank the reviewer for giving us the opportunity to clarify our results. We counted the number of eggs laid per animal from day 1 to day 4 of adulthood (Rebuttal Fig. 1). As expected, laid eggs did not hatch in the presence of FUdR in ND and HGD-1 animals. Overall, in the absence of FUdR, the number of

Rebuttal Figure 1. Total number of eggs laid per ND and HGD-1 animals at day 1, 2, 3, and 4 of adulthood in the absence (-) and presence (+) of FUdR ($n=10$).

laid eggs in HGD-1 was like ND animals during the 4 days of adulthood. In contrast, the number of laid eggs is dramatically reduced in HGD-1 animals compared to HD in the presence of FUdR. We also observed a higher number of unlaied eggs in HGD-1 animals in the presence of FUdR (data not shown). However, the number of laid eggs in HGD-1 remained relatively low through the reproductive period of the animal in the presence of FUdR. We have modified the manuscript and the figure for clarity.

Updated manuscript (lines 228-232)

“The number of eggs laid per HGD-1 animal was slightly reduced compared to ND animals in the absence of FUdR at day 4 of adulthood (Supplementary Fig. 3e). Interestingly in the presence of FUdR, the number of eggs laid per HGD-1 animal was dramatically reduced compared to ND animals. Together with the lifespan assays in the absence or presence of FUdR, findings suggest that HGD in young animals decreases both the number of laid eggs and lifespan.”

New Supplementary Figure S3e:

9. On Line 219 it is written “findings suggest that embryogenesis counteracts life extension induced by HGD in aged animals.” No brood size experiments are shown for HGD-5 animals, just on ND and HGD-1 animals. Therefore, this statement is inaccurate and could be changed to “findings suggest that HGD in young animals decreases both the brood size and lifespan”.

We have modified the statement according to the reviewer suggestion.

10. Line 330 It has already been shown that calorie restriction when initiated later in life can extend lifespan of several organisms including mice (although not to the extent it can when it is initiated earlier).

We understand that our statement was oversimplistic and the literature about the role dietary restriction and ageing is extensive. Therefore, we deleted the sentence “However, it remains to be determined whether or not calorie restriction initiated at the post-reproductive stage will result in lifespan extension, as shown in young animals” to keep the discussion focused on HGD, cellular stress, and ageing.

Minor comments: wording

We thank the reviewers for taking the time to report these mistakes. We have made the changes accordingly for these listed below.

- Line 197: by -> starting on – modified
- Line 199: promoting -> regulating – modified
- Line 209: promote -> promotes – modified
- Line 214: days 7 -> day 7 – modified
- Line 241: from extending -> from the nucleus and is not involved in extending – modified
- Line 257: are -> are likely – modified
- Line 268: renders -> plays – modified
- Line 276: of IRE-1 -> of the IRE-1 – modified
- Line 290: suggest -> suggests – modified
- Line 322: sapsins -> saposins [perhaps I don’t know what a sapsin is] – modified
- Line 329: combined -> when combined – modified
- Line 337: calorie restriction increases -> some dietary restriction protocols increase – modified
- Line 341 and 342: I would remove the words “to extend the lifespan of aged animals” [unfounded speculation] – modified
- Line 346: animal -> animals – modified
- Line 361: fitness -> relative function – modified

Line 366: ATF-6 and PEK-1 while attenuating IRE-1 pathways -> ATF-6 and PEK-1 pathways while attenuating the IRE-1 pathway – modified

Line 368: in preventing -> to prevent – modified

Reviewers' Comments:

Reviewer #4:

Remarks to the Author:

The authors adequately responded to my comments from the first round of review. Therefore, all of the data now appears appropriate. However, the wording is now unclear in many places throughout the paper and suggested improvements are shown below.

Minor wording changes suggested:

Title: Since glucose is not the entire diet but just an addition to the E. coli diet, I would suggest changing the title to "The unfolded protein response modulates lifespan in chemically-sterilized C. elegans upon glucose addition" or better "The unfolded protein response alters glucose modulation of lifespan in chemically-sterilized C. elegans" or perhaps even better "The unfolded protein response plays a role in reversing the effects of glucose on lifespan in chemically-sterilized C. elegans".

Line 63: with high glucose diet -> supplemented with high glucose

Line 86: by adapting to stress conditions and restoring -> to adapt to stress conditions to restore

Line 88: UPR -> The UPR

Line 99: intolerance -> susceptibility

Line 113: lifespan -> decreased lifespan

Line 126: significantly extended their -> had a significantly extended

Line 140: adulthood -> adulthood and found an increased rate of body bending

Line 165: thereafter needs -> needs thereafter

Line 167: metabolisms -> metabolism

Line 176: identified and demonstrated genes -> examined metabolites

Line 176: in which -> and found that

Line 177: the most -> more

Line 179: Myristic -> Levels of myristic

Line 183: increase of the -> increased number of

Line 184: organelle, namely lipid droplet, -> organelles, namely lipid droplets,

Line 184: stored the highest -> contained the highest number

Line 206: stem cell population which is precursor -> stem cells, which are precursors

Line 207: normal diet -> ND

Line 238: induces -> increases the levels of

Line 246: question -> determine

Line 258: by adapting to stress conditions and restoring -> to adapt to stress conditions and restore

Line 291: splices -> activation causes increased splicing of

Lines 301 and 310: lifespan -> decreased lifespan

Line 309: Conjointly -> Together

Line 318: germ-cell loss C. elegans mutant glp-1(lof) -> C. elegans glp-1(lof) mutant that shows a complete loss of germ cells

Line 330: damages -> damage

Line 340: hypothesise -> hypothesize

Line 351: fitness of -> ability to activate

Line 351: during ageing -> until the fifth day of adulthood

Line 352: impartial -> not involved

Line 360: Protein quality control is imperative in -> Preventing decreased protein quality control is imperative for

Line 367: Remove the word "mRNA"

Line 367: xbp-1 -> XBP-1

Line 376: animal -> animals

Line 377: attenuating -> attenuating the

REPLY TO THE REVIEWER

Reviewer #4:

The authors adequately responded to my comments from the first round of review. Therefore, all of the data now appears appropriate. However, the wording is now unclear in many places throughout the paper and suggested improvements are shown below.

Minor wording changes suggested:

Title: Since glucose is not the entire diet but just an addition to the *E. coli* diet, I would suggest changing the title to “The unfolded protein response modulates lifespan in chemically-sterilized *C. elegans* upon glucose addition” or better “The unfolded protein response alters glucose modulation of lifespan in chemically-sterilized *C. elegans*” or perhaps even better “The unfolded protein response plays a role in reversing the effects of glucose on lifespan in chemically-sterilized *C. elegans*”.

We think the title “The unfolded protein response plays a role in reversing the effects of glucose on lifespan in chemically-sterilized *C. elegans*” is the most appropriate. To comply with the journal formatting instructions, the title was reduced to 15 words as follow: “The unfolded protein response reverses the effects of glucose on lifespan in chemically-sterilized *C. elegans*”.

Changes below are highlighted in red of the revised manuscript.

Line 63: with high glucose diet -> supplemented with high glucose **Modified**
Line 86: by adapting to stress conditions and restoring -> to adapt to stress conditions to restore **Modified**
Line 88: UPR -> The UPR **Modified**
Line 99: intolerance -> susceptibility **Modified**
Line 113: lifespan -> decreased lifespan **Modified**
Line 126: significantly extended their -> had a significantly extended **Modified**
Line 140: adulthood -> adulthood and found an increased rate of body bending **Modified**
Line 165: thereafter needs -> needs thereafter **Modified**
Line 167: metabolisms -> metabolism **Modified**
Line 176: identified and demonstrated genes -> examined metabolites **Modified**
Line 176: in which -> and found that **Modified**
Line 177: the most -> more **Modified**
Line 179: Myristic -> Levels of myristic **Modified**
Line 183: increase of the -> increased number of **Modified**
Line 184: organelle, namely lipid droplet, -> organelles, namely lipid droplets, **Modified**
Line 184: stored the highest -> contained the highest number **Modified**
Line 206: stem cell population which is precursor -> stem cells, which are precursors **Modified**
Line 207: normal diet -> ND **Modified**
Line 238: induces -> increases the levels of **Modified**
Line 246: question -> determine **Modified**
Line 258: by adapting to stress conditions and restoring -> to adapt to stress conditions and restore **Modified**
Line 291: splices -> activation causes increased splicing of **Modified**
Lines 301 and 310: lifespan -> decreased lifespan **Modified**
Line 309: Conjointly -> Together **Modified**
Line 318: germ-cell loss *C. elegans* mutant *glp-1(lof)* -> *C. elegans glp-1(lof)* mutant that shows a complete loss of germ cells **Modified**
Line 330: damages -> damage **Modified**
Line 340: hypothesise -> hypothesize **Modified**
Line 351: fitness of -> ability to activate **Modified**
Line 351: during ageing -> until the fifth day of adulthood **Modified**
Line 352: impartial -> not involved **Modified**
Line 360: Protein quality control is imperative in -> Preventing decreased protein quality control is imperative for **Modified**
Line 367: Remove the word “mRNA” **Modified**
Line 367: *xbp-1* -> XBP-1 **Modified**
Line 376: animal -> animals **Modified**
Line 377: attenuating -> attenuating the **Modified**